# A Kernel Two-sample Test for Dynamical Systems

## Abstract

Evaluating whether data streams are drawn from the same distribution is at the heart of various machine learning problems. This is particularly relevant for data generated by dynamical systems since such systems are essential for many real-world processes in biomedical, economic, or engineering systems. While kernel two-sample tests are powerful for comparing independent and identically distributed random variables, no established method exists for comparing dynamical systems. The main problem is the inherently violated independence assumption. We propose a two-sample test for dynamical systems by addressing three core challenges: we (i) introduce a novel notion of mixing that captures autocorrelations in a relevant metric, (ii) propose an efficient way to estimate the speed of mixing relying purely on data, and (iii) integrate these into established kernel two-sample tests. The result is a data-driven method that is straightforward to use in practice and comes with sound theoretical guarantees. In an example application to anomaly detection from human walking data, we show that the test is readily applicable without any human expert knowledge and feature engineering.

## 1 Introduction

We consider the two-sample problem of determining whether two distributions are different. In particular, we generalize the well-established kernel two-sample test (Gretton et al., 2012a) to dynamical systems and stochastic processes with certain mixing properties, which we make precise in this paper.

The kernel two-sample test approximates a metric on the space of probability distributions, the maximum mean discrepancy (MMD), through kernel-based techniques. Due to its powerful theoretical properties and versatile applicability, kernel two-sample testing is a prominent method in the machine learning community (Long et al., 2017; Tolstikhin et al., 2018; Muandet et al., 2017; Schölkopf & Smola, 2001). While we can exploit parts of existing kernel-based results, and especially their theoretical guarantees, the extension to comparing dynamical systems is not straightforward. This is mainly because kernel two-sample testing was initially developed for independent and identically distributed (i.i.d.) random variables (Gretton et al., 2012a). The i.i.d. assumption in the test is critical, but it is violated by the very nature of dynamical systems: through the dynamics, samples are coupled to past samples. To address this issue, we introduce a novel notion of mixing that considers the dependence of data through time with respect to the MMD. Intuitively, mixing reveals how fast autocorrelations decay and, thus, how long we need to wait in-between samples for data to be (approximately) independent. Our new mixing notion can be efficiently estimated from data and is particularly synergistic with kernel two-sample tests since both measure distances of probability distributions with respect to the same metric—the MMD. By estimating the decay of dependency and embedding it into well-established algorithms, we obtain a powerful test for comparing dynamical systems.

Mathematical literature often distinguishes explicitly between deterministic dynamical systems and stochastic processes. In particular, establishing mixing properties for deterministic dynamical systems is an extremely challenging problem and constructing examples that are provably mixing is hard. Further, common mixing properties that are used for stochastic systems are too restrictive and not applicable to deterministic systems (Hang et al., 2017). We propose mixing in MMD, which is applicable to both classes of problems—stochastic and deterministic systems. Further, we show that mixing in MMD is even less restrictive than certain deterministic mixing types ($\mathcal{C}$-mixing Hang et al. (2018)). For suitable choices of kernels and function

spaces $\mathcal{C}$, we can show that $\mathcal{C}$-mixing implies MMD-mixing. Based on standard examples with well-established $\mathcal{C}$-mixing properties (the $\beta$-map, logistic map, and Gauss map), we demonstrate empirically that they are indeed mixing in MMD. Additionally, we also consider mixing properties of chaotic and stochastic systems and further, also raw sensor data from human walking experiments.

Despite their practical relevance, there is no established data-driven way of comparing dynamical systems. For biomedical systems such as the human cardiovascular system, central nervous system, or musculoskeletal system, implementing a principled comparison of systems based on their output sequences in different time intervals can help to detect diseases or quantify their severity. For example, alterations or unusual patterns in human gait can be indicators for early stages of Parkinson's disease (Pistacchi et al., 2017). An algorithm that automatically detects such alterations by comparing new data to labeled records could help physicians in their decision-making. Current state-of-the-art solutions rely on manually engineered and selected features and thus require expert knowledge (Nguyen et al., 2019). Similarly, feature-based solutions have been proposed for electro-myography-based detection of spasticity (Misgeld et al., 2015; Lueken et al., 2015). But clearly, the success of such approaches critically depends on the expressiveness of these features and on how well the problem is understood.

Modern engineering applications are another prominent and relevant example. They often leverage computer simulations instead of directly interacting with the physical plant since real experiments are more expensive, time-consuming, and cause wear on the hardware. Besides, being able to predict the response of a physical plant based on a mathematical model enables powerful learning algorithms (Hwangbo et al., 2019), model-predictive control (Qin & Badgwell, 2003), and digital twins in future manufacturing (Jeschke et al., 2017). The success of these methods, however, is critically intertwined with the model accuracy. Thus, it is essential to ensure accurate models, for example, by comparing data generated from the simulation model with data collected from the real system.

By combining mixing properties with kernel-based techniques, we obtain a powerful statistical test for comparing dynamical systems. We demonstrate the efficiency and robustness of the proposed test numerically and on experimental data. In particular, we consider human walking experiments and analyze raw data from an inertial measurement unit (IMU) to detect anomalies in the walking pattern. Without the need for human expert knowledge or fitting model parameters, our test outperforms standard baselines in deciding which of the trajectories were generated with an attached knee orthosis, which restricts the movement of the joint.

**Contributions:** We propose a kernel two-sample test for dynamical systems. By developing a new notion of mixing that can be estimated from data, we generalize powerful theoretical guarantees from the i.i.d. setting to certain dynamical systems. The derived method is straightforward to use, and well-established implementations of the kernel two-sample test can be leveraged. We demonstrate the robustness and efficiency of the method on real-world data, where we achieve better results than standard baselines, without relying on feature engineering or expert knowledge. Code and data will be made available upon publication.

## 2 Related Work

There is only limited literature that explicitly investigates the question of how to compare dynamical systems. One possibility is the embedding of dynamical systems as infinite-dimensional objects into reproducing kernel Hilbert spaces (RKHS) with specifically designed kernels such as Binet-Cauchy kernels (Vishwanathan et al., 2007) or generalizations thereof as proposed in Ishikawa et al. (2018) and Ishikawa et al. (2019). A similar function-analytical approach is considered in Mezic (2016) and Klus et al. (2020), where the authors consider Koopman and Perron-Frobenius operators to obtain linear dynamics in an infinite-dimensional space. These articles leverage specifically designed kernels and linear operators associated with dynamical systems to obtain an embedding. However, none of the above articles proposes a statistical test that compares dynamical systems. This would require further finite sample and error bounds on the approximations of the infinite-dimensional operators, which is non-trivial. Our approach leverages concentration results that synergize well with kernel-based techniques and in particular, kernel mean embeddings.

The critical technical issue for dealing with dynamical systems is non-i.i.d. data. There are several extensions of kernel two-sample tests that have been developed (Zaremba et al., 2013; Gretton et al., 2012b; Doran et al., 2014; Lloyd & Ghahramani, 2015; Chwialkowski et al., 2014; Chwialkowski & Gretton, 2014) to make them applicable to a broader range of problems, where non-i.i.d. data is also an issue. However, the strong mixing properties that are typically postulated limit the applicability of the results to dynamical systems. Surprisingly, there is only very limited work that addresses the estimation of mixing coefficients from data, as also acknowledged and emphasized in (McDonald et al., 2011). The approach proposed in (McDonald et al., 2011) is different from our work, as mixing is considered with respect to the total variation norm, which requires the estimation of complex intermediate objects, whereas we estimate mixing properties directly from data. We propose a new mixing notion that synergizes well with kernel two-sample tests and that can also be *estimated* from data (in contrast to most other mixing notions). A similar idea of mixing in RKHS, has very recently been introduced in (Chérief-Abdellatif & Alquier, 2022). The paper, however, focuses on parameter estimation with respect to minimizing the MMD as a loss function. Further, the precise notion of mixing differs from ours. It is shown that certain types of systems satisfy their notion of mixing, however, that work does not estimate mixing from data. In (Wynne & Duncan, 2022), another recent approach is presented. In essence, the paper investigates whether two samples of functions have the same underlying distribution. Functional data is directly embedded into an RKHS by extending the theory to kernels that live on function spaces. The elegant kernel design might be a useful for extensions of our work, where multiple correlated joint distributions need be compared. Right now, we compare stationary distributions and data is decorrelated via mixing.

The problem of comparing dynamical systems is also present in control theory and was, for example, recently studied in Umlauft & Hirche (2019); Solowjow & Trimpe (2020) by considering the question of when to trigger model updates. In robust control, there is the notion of the gap metric (Zhou & Doyle, 1998), which compares the closed-loop behavior of dynamical systems. These approaches are particularly promising to quantify the similarities between dynamical systems when trying to achieve effective transfer learning, as shown in (Sorocky et al., 2020). However, they usually rely on a given model or a certain linear structure in the system. But estimating such models of nonlinear systems can be difficult in practice (Schoukens & Ljung, 2019; Schön et al., 2011; Brunton et al., 2017; Ljung, 2001). Similarly, estimating the stationary measure of a dynamical system is also a highly non-trivial problem (Hang et al., 2018; Luzzatto et al., 2005). In our approach, we do not require any intermediate objects such as the dynamics, density function, or noise models. Instead, we compare stationary distributions of dynamical systems directly from data.

## 3 Assumptions and Problem Formulation

In the following, we introduce the mathematical objects that we will consider in this paper. Afterward, we make the problem precise.

### 3.1 Stationary, Ergodic, and Mixing Systems

Let $(\Omega, \mathcal{A}, P)$ be a probability space, $S \subset \mathbb{R}^d$ a compact set, which is the state space of the dynamical system, and $\mathcal{B}$ the corresponding Borel $\sigma$-algebra. We define a stochastic dynamical system or stochastic process as a collection of random variables $\{X_k\}$ indexed in discrete time $k \in \mathbb{N}$ and $X_k \colon \Omega \to S$. Next, we introduce some required properties of the process.

**Definition 1 (Stationary)** *A system is stationary if the joint distribution of its states is time-invariant.*

In addition to stationary behavior, we also require ergodicity. While stationarity ensures time-invariant distributions, ergodicity guarantees that the statistical properties of the system do not differ over multiple realizations. We use a standard definition that goes back to Birkhoff (1931).

**Definition 2 (Ergodic)** *Assume the system $\{X_k\}$ is stationary with distribution $\mathbb{P}$. We call the system ergodic if for all $f \in L_{\mathbb{P}}^1(S)$ and $\mathbb{P}$-almost all initial states we have*

$$\lim_{N \to \infty} \frac{1}{N} \sum_{k=0}^{N-1} f(X_k) = \int_S f(y) \mathrm{d}\mathbb{P}(y) \quad a.s.. \tag{1}$$

Equation equation 1 is in some sense a realization of the law of large numbers, and both sides of the equation yield the expected value $\mathbb{E}_{X \sim \mathbb{P}}[f(X)]$. In particular, it allows us to estimate $\mathbb{E}[X_k]$ (the distribution is invariant for all $k$) from long enough sample paths. Different types of convergence and test functions in eq. equation 1 yield more sophisticated ergodic theorems. Nonetheless, there can still be severe autocorrelations and if $X_k$ is known, this may have a drastic impact on the distribution of $X_{k+1}$. Thus, we require additional mixing assumptions.

Classically, mixing is introduced in terms of dependencies between $\sigma$-algebras and intuitively, deals with the autocorrelations in the system. Here, we consider a covariance-based approach to mixing, which is more useful and convenient for us since there is a natural connection to Hilbert-Schmidt theory in RKHSs. Both approaches are introduced in Bradley et al. (1987). We begin with a general definition based on (Bradley et al., 1987, eq. (1.2)) and tailor it to our problem afterward.

**Definition 3 (Measure of Dependence)** *Assume $\mathcal{F}$ and $\mathcal{G}$ are suitable function spaces. The measure of dependence is defined as*

$$\sup_{f \in \mathcal{F}, g \in \mathcal{G}} \frac{|\mathbb{E}[fg] - \mathbb{E}[f]\mathbb{E}[g]|}{\|f\|_p \|g\|_q}, \tag{2}$$

*where $p$ and $q$ are Hölder pairs.*

A possible choice for the function spaces is $\mathcal{F} = \mathcal{G} = L^2$, which is referred to as strong mixing when eq. equation 2 converges to zero and naturally implies ergodicity in $L^2$ when considering $f = g$. There are various valid choices and many are discussed in (Bradley, 2005; Hang et al., 2017).

Here, we propose to consider unit balls in reproducing kernel Hilbert spaces for $\mathcal{F}$ and $\mathcal{G}$, which has to the best of our knowledge not been done before.

**Definition 4 (Mixing)** *Assume $\mathcal{F}$ and $\mathcal{G}$ are unit balls in the same RKHS. We call a system mixing if*

$$\sup_{f \in \mathcal{F}, g \in \mathcal{G}} \mathrm{Cov}(f(X_t), g(X_{t+a})) \to 0 \quad \text{for} \quad a \to \infty. \tag{3}$$

Later, we will investigate this property in more detail and leverage powerful estimators in form of the Hilbert-Schmidt independence criterion to determine mixing properties. Estimators for the speed of mixing are usually a critical issue when working with mixing arguments. In related work, the speed and type of mixing is almost exclusively postulated. In contrast, we test if a process is mixing and estimate the actual speed.

An important special case that we will investigate in detail are state-space models or Markov chains with continuous state spaces of the type $X_{k+1} = \phi(X_k) + \epsilon_k$, where $\phi$ is an appropriate dynamics function and $\epsilon_k$ the process noise. This system description is highly relevant in systems and control theory and more recently, reinforcement learning. Further, we will also consider chaotic systems, where $\epsilon_k \equiv 0$. These are deterministic and violate common probabilistic mixing assumptions.

### 3.2 Problem Formulation

Consider two stationary and mixing (cf. def. 4) systems $\{X_k\}$ and $\{Y_k\}$ with stationary distributions $\mathbb{P}_X$ and $\mathbb{P}_Y$. We want to decide whether $\{X_k\}$ and $\{Y_k\}$ are different based on the data streams $X = \{X_0, X_1, \ldots, X_n\}$ and $Y = \{Y_0, Y_1, \ldots, Y_n\}$. We assume $X_k, Y_k \in S$ and, in general, $X_0 \neq Y_0$. Further, we assume that the dynamical systems have converged to their stationary distribution.

We propose to compare dynamical systems by testing whether their stationary probability measures coincide. Thus, we obtain the null hypothesis

$$H_0 : \mathbb{P}_X = \mathbb{P}_Y, \tag{4}$$

which we try to reject with high confidence. For our method, it is not necessary to estimate or construct any intermediate objects such as the dynamics function $f$, nor the measures $\mathbb{P}_X$ and $\mathbb{P}_Y$.

The main challenge lies in coping with the autocorrelations within the data streams. These autocorrelations are critical and void commonly used concentration results, such as the famous Hoeffding's or McDiarmid's inequalities.

We consider a two-sample setting between two data streams $X$ and $Y$. However, this can easily be applied to settings where we want to investigate whether a given model coincides with reality. Then, samples obtained through sensor measurements can be compared with samples generated by simulating a given model.

## 4 Technical Preliminaries

The main idea of this paper can be summarized as generalizing kernel two-sample tests (Gretton et al., 2012a) to dynamical systems through a data-based mixing approach. Essentially, we propose to wait *long enough* between consecutive samples. Quantifying how long to wait to enforce negligibly small autocorrelations is the core question, which is addressed in sec. 5.2. In the numerical section, we construct linear systems with arbitrary slow mixing properties. We begin by summarizing key results from kernel two-sample tests and kernel mean embeddings.

### 4.1 Kernel Two-sample Test

An elegant and efficient comparison of probability distributions can be achieved with kernel two-sample tests (Gretton et al., 2012a). The distributions are embedded into an RKHS, where it becomes tractable to compute certain metrics on the space of probability distributions such as the MMD. The following definitions and theorems are taken from Gretton et al. (2012a).

**Definition 5 (MMD)** *Let $(S, d)$ be a metric space and let $\mathbb{P}_X, \mathbb{P}_Y$ be two Borel probability measures defined on $S$. Further, let $\mathcal{F}$ be the unit ball in an RKHS on $S$. We define the maximum mean discrepancy by*

$$\mathrm{MMD}^2[\mathbb{P}_X, \mathbb{P}_Y] = \sup_{g \in \mathcal{F}} (\mathbb{E}_{\mathbb{P}_X}[g] - \mathbb{E}_{\mathbb{P}_Y}[g])^2. \tag{5}$$

The MMD yields a semi-metric between probability distributions and can be efficiently estimated by embedding the distributions into an RKHS $\mathcal{H}$ with the aid of kernel mean embeddings (Muandet et al., 2017). It is a challenging problem to compute equation 5 directly since $\mathcal{F}$ is usually infinite-dimensional. However, by kernelizing it, we can estimate equation 5 from data.

**Theorem 1** *Assume $k$ is a kernel and $\mathcal{F}$ is again the unit ball in the corresponding RKHS $\mathcal{H}$. Further, assume $(X_1, \ldots, X_n)$ and $(Y_1, \ldots, Y_m)$ are drawn i.i.d. from $\mathbb{P}_X$ and $\mathbb{P}_Y$, respectively. Then, a biased estimate of equation 5 is given by*

$$\mathrm{MMD}_b^2[X, Y] = \frac{1}{n^2} \sum_{i,j=1}^{n} k(X_i, X_j) + \frac{1}{m^2} \sum_{i,j=1}^{m} k(Y_i, Y_j) - \frac{2}{mn} \sum_{i=1}^{n} \sum_{j=1}^{m} k(X_i, Y_j). \tag{6}$$

The additional requirement of a characteristic kernel ensures that the embedding of the probability distribution is injective and, thus, a metric is obtained. The kernel $k$ can, for example, be chosen as a Gaussian kernel since it is well known to be characteristic (Gretton et al., 2012a).

**Theorem 2** *Assume $k$ is a characterstic kernel and $\mathcal{F}$ is the unit ball in the corresponding RKHS $\mathcal{H}$. Then $\mathrm{MMD}^2[\mathbb{P}_X, \mathbb{P}_Y] = 0$ if, and only if, $\mathbb{P}_X = \mathbb{P}_Y$.*

Essentially, we do not require any prior knowledge or parameterization of $\mathbb{P}_X$ and $\mathbb{P}_Y$. Access to i.i.d. samples from these distributions is sufficient. In practice, however, we only have access to finitely many data points and, thus, receive an estimate of the MMD from equation 6. This estimate is expected to have some deviation, i.e., even for identical distributions, the test statistic will be larger than zero. Therefore, we need finite sample bounds that quantify the convergence speed of the empirical MMD to obtain confidence bounds. Gretton et al. (2012a) introduce several such bounds of the type

$$\mathbb{P}\left[|\mathrm{MMD}_b[X,Y] - \mathrm{MMD}[\mathbb{P}_X,\mathbb{P}_Y]| \geq \kappa(\alpha,n)\right] \leq \alpha. \tag{7}$$

Under the null hypothesis $\mathbb{P}_X = \mathbb{P}_Y$, we can obtain the rejection region $\mathrm{MMD}_b[X,Y] \geq \kappa(\alpha,n) = \sqrt{2\frac{K}{n}}(1 + \sqrt{2\log\alpha^{-1}})$ for a test with level $\alpha$, where $K$ is the supremum of the kernel (Gretton et al., 2012a, Corollary 9). However, these results rely on the independence assumption and, hence, cannot be used for comparing dynamical systems.

## 4.2 Hilbert-Schmidt Independence Criterion

The Hilbert-Schmidt independence criterion (HSIC) (Gretton et al., 2008) quantifies dependence between random variables. Generally, two random variables $X$ and $Y$ are independent if their joint distribution factorize, i.e., $\mathbb{P}_{X,Y} = \mathbb{P}_X \otimes \mathbb{P}_Y$, where $\otimes$ denotes the tensor product. Estimating the involved objects from data is usually intractable. Instead, the difference in MMD can elegantly be expressed through the HSIC.

**Definition 6 (HSIC, Sejdinovic et al. (2013, Def. 11))** *Let $X \sim P_X$ and $Y \sim P_Y$ be random variables with joint distribution $P_{X,Y}$. The HSIC is defined as*

$$\mathrm{HSIC}(X,Y) = \mathrm{MMD}_{\mathcal{H}\otimes\mathcal{H}}[P_X \otimes P_Y, P_{X,Y}]. \tag{8}$$

Similar to the kernel two-sample test, it is possible to express equation 8 in terms of kernel evaluations. Further, it is also possible to provide high confidence bounds and thus, obtain an efficient statistical test.

As the name suggests, the HSIC is closely related to Hilbert-Schmidt operators. These well-behaved operators are well investigated in functional analysis and in general, are bounded operators between Hilbert spaces. Further, the space of Hilbert-Schmidt operators between two reproducing kernel Hilbert spaces $\mathcal{H}$ and $\mathcal{G}$ forms itself a Hilbert space, which is isomorphic to the product space $\mathcal{H} \otimes \mathcal{G}$ given by the product kernel (Muandet et al., 2017, Page 35).

Here, we want to emphasize the connection between the HSIC and the covariance operator $\mathcal{C}_{XY}$ in terms of the Hilbert-Schmidt norm (Muandet et al., 2017, Eq. 3.37)

$$\|\mathcal{C}_{X,Y}\|_{\mathrm{HS}} = \mathrm{HSIC}(X,Y) \tag{9}$$

and the representation of $\mathcal{C}_{X,Y}$ as the unique bounded operator that satisfies the property

$$\langle g, \mathcal{C}_{X,Y}f \rangle_{\mathcal{G}} = \mathrm{Cov}[g(Y), f(X)] \tag{10}$$

for all $g \in \mathcal{G}$ and $f \in \mathcal{H}$. Equivalently, the covariance operator can also be defined in terms of tensor spaces (Muandet et al., 2017, Sec. 3.2), however, equation 10 connects nicely to standard mixing expressions (cf. eq. equation 3). Further, the HSIC framework provides rich results such as efficient estimators and concentration results.

The general framework is highly flexible and can deal with a variety of objects. For our problem, we can use a simplified setting, where both systems belong to the same space, which is consistent with the setup for the kernel two-sample test. Estimations of equation 8 in terms of kernel evaluations can be found in (Gretton et al., 2008, Equation (4)).

## 4.3 Joint Independence—dHSIC

Pfister et al. (2018) extended the HSIC to $d$-dimensional random vectors and thus, investigate

$$\mathrm{dHSIC}(X) = \mathrm{MMD}_{\otimes_{i=1}^d \mathcal{H}}[P_{X_1} \otimes \ldots \otimes P_{X_d}, P_{X_1,\ldots,X_d}]. \tag{11}$$

Similarly as for the kernel two-sample test and the classical HSIC independence test, it is possible to quantify the convergence speed and thus, obtaining a threshold $\kappa(\alpha, n)$ for statistical testing. Intuitively, this should be the independence notion that we need for the kernel two-sample test. However, we use a slightly different property, which we introduce in def. 9.

## 5 MMD-Mixing

In this section, we will focus on data from one system $\{X_k\}$ and investigate the temporal dependencies. We assume access to multiple independent trajectories, which we indicate through superscripts $\{X_k^{(i)}\}$. Due to the ergodicity and stationarity assumptions, we obtain a well-defined underlying distribution $\mathbb{P}$ for which we can test. Next, we will introduce the new concept of MMD-mixing that quantifies the decay of autocorrelations with respect to the MMD and connect back to the HSIC.

### 5.1 Time Shifts and MMD-mixing

Let the trajectory $X_0, X_1, \ldots, X_n$ be subject to a given sampling rate. Generally, autocorrelations decay over time, and far apart samples are approximately independent if the underlying system is mixing. Hence, we propose to increase the time between consecutive samples to reduce dependencies. The slower sampling rate is denoted through the time shift $a \in \mathbb{N}$ and yields data $X_0, X_a, X_{2a}, \ldots, X_{an}$. Essentially, the question is how to determine and estimate $a$ to ensure approximately independent data points $X_0, X_a, X_{2a}, \ldots, X_{an}$. We begin with a simplified setting and assume a sample from the stationary measure $X_0 \sim \mathbb{P}_{X_0} = \mathbb{P}$.

**Definition 7 (MMD-mixing)** *We call a process* MMD-mixing *if*

$$\mathrm{MMD}_{\mathcal{H} \otimes \mathcal{H}}[\mathbb{P}_{X_0} \otimes \mathbb{P}_{X_a}, \mathbb{P}_{X_0, X_a}] \to 0 \quad \text{for } a \to \infty. \tag{12}$$

This definition only considers the distributions at two points in time. Due to the stationarity of the system, we can move the timeshift through time and also consider different pairs in time. Due to the connection to Hilbert-Schmidt theory and covariance operators, it is also possible to consider expressions similar to equation 10.

**Proposition 1** *Let* $\{X_k\}$ *be an MMD-mixing process. Then,*

$$\mathrm{HSIC}(X_0, X_a) \to 0 \quad \text{for } a \to \infty. \tag{13}$$

We assume that the underlying kernel $k$ is characteristic and refer to it as the base kernel. Further, we assume to have access to $m$ independent trajectories. In this setting, we can readily apply the HSIC equation 8 framework. In particular, we pick two points in time from each trajectory, $X_0^{(i)}$ and $X_a^{(i)}$. Then, we divide the data into $X_0 = \{X_0^{(1)}, X_0^{(2)}, \ldots, X_0^{(m)}\}$ and $X_a = \{X_a^{(1)}, X_a^{(2)}, \ldots, X_a^{(m)}\}$. The sets are, per construction, i.i.d. within themselves since the $m$ trajectories are independent and due to the additional stationarity and ergodicity assumptions. Next, we can compute $\mathrm{HSIC}(X_0, X_a)$ and iteratively increase $a$. If we pick $a$ large enough, the HSIC will eventually become arbitrarily small. MMD-mixing ensures that $\mathrm{HSIC}(X_s, X_{s+a}) \to 0$ for $a \to \infty$. For practical algorithms, we will fix a small $\epsilon > 0$ and enforce $\mathrm{HSIC}(X_s, X_{s+a^*}) < \epsilon$. In our experiments, we pick $\epsilon$ as the standard test threshold of the HSIC (cf. fig. 3).

### 5.2 Extended MMD-mixing

Next, we extend our arguments to subtrajectories instead of considering two single points. Similarly as before, we use the notation $\mathbb{P}_{X_0, \ldots, X_s}$ and $\mathbb{P}_{X_{s+a}, \ldots, X_{2s+a}}$ for the distributions of the subtrajectories of length $s$ (which is the joint distribution over the first $s$ states) and $\mathbb{P}_{(X_0, \ldots, X_s),(X_{s+a}, \ldots, X_{2s+a})}$ for the joint distribution of the subtrajectories.

**Definition 8 (Extended MMD-mixing)** *We call a process* extended MMD-mixing *if*

$$\mathrm{MMD}_{(\otimes_{i=1}^s \mathcal{H}) \otimes (\otimes_{i=1}^s \mathcal{H})}[\mathbb{P}_{X_0, \ldots, X_s} \otimes \mathbb{P}_{X_{s+a}, \ldots, X_{2s+a}}, \mathbb{P}_{(X_0, \ldots, X_s),(X_{s+a}, \ldots, X_{2s+a})}] \to 0 \quad \text{for } a \to \infty. \tag{14}$$

Clearly, we require an appropriate kernel to extend the MMD to joint distributions. In particular, tensor products of the base kernel need to be strong enough to distinguish the joint distributions. Szabó & Sriperumbudur (2018) discuss various tensor constructions, which we will leverage here.

**Lemma 1 (Choice of Kernel I)** *Let $k$ be a characteristic kernel. Then $k^s = \otimes_{i=1}^s k$ is also characteristic.*

**Proof:** The statement follows directly from Szabó & Sriperumbudur (2018, Theorem 4), which considers a more general problem setting. $\square$

Due to the tensor construction, we naturally obtain MMD-mixing with respect to $k$ as introduced in Definition 7, for a process that is extended MMD-mixing with respect to $k^s$.

### 5.3 Joint Independence

To apply the mixing results to kernel two-sample testing, we require one more step. We need joint independence between all samples. Intuitively, this coincides with the dHSIC framework (cf. equation 11) and can be implemented through more sophisticated tensor kernels that embed multiple data points or subtrajectories simultaneously.

**Lemma 2 (Choice of Kernel II)** *Assume $k^s$ is a characteristic kernel. Then the tensor kernel $k^{s,n} = \otimes_{i=1}^n k^s$ is an $\mathcal{I}$-characteristic kernel, which makes the kernel suitable for joint independence testing.*

**Proof:** Follows from (Szabó & Sriperumbudur, 2018, Theorem 4). $\square$

To combine mixing with kernel two-sample testing, we require the following technical assumption.

**Definition 9 (Approximately $\epsilon$-independent)** *Let $\{X_k\}$ be an MMD-mixing process. We call data $X = X_{a^*}, X_{2a^*}, \ldots, X_{na^*}$ approximately $\epsilon$-independent if there is a time shift $a^*$ and threshold $\kappa(\epsilon, n)$ that yields*

$$\mathbb{P}[\mathrm{MMD}_b(X, \bar{X}) \geq \kappa] < \epsilon, \tag{15}$$

*where $\bar{X}$ is data that has been sampled independently from the stationary distribution $\mathbb{P}$.*

An important technical detail here is the fact that we consider the MMD with respect to the kernel $k$ and not the tensor kernel $k^{s,n}$. Independent and identically distributed data naturally satisfies the above property when $\epsilon = \alpha$ and $\kappa$ is chosen correspondingly for a level-$\alpha$ kernel two-sample test. In practice, we apply a level $\epsilon$ HSIC test to multiple independent trajectories in order to determine an $a^*$, which satisfies equation 15.

### 5.4 Connections to Other Mixing Notions

There are various types of mixing that essentially all describe the decay of autocorrelations. An extensive discussion of the relationship between different measures of dependencies can be found in Bradley (2005). The importance of covariance-based expressions for mixing is utilized in Bradley et al. (1987) to investigate how they can dominate each other.

Mixing properties are notoriously difficult or even impossible to estimate, and many types of mixing do not apply to large classes of dynamical systems (Hang et al., 2017). Our proposed type of mixing can be estimated from data and yields advantageous theoretical properties. In McDonald et al. (2011), the $\beta$-mixing coefficient is estimated through involved density estimations. While the authors emphasize that they solve a more difficult problem to obtain a solution to a simpler one, this is still one of the few existing approaches to estimate the speed of mixing.

We start with defining the $\beta$-mixing coefficient as in (McDonald et al., 2011):

$$\beta(a) = \sup_s \|\mathbb{P}_{-\infty}^s \otimes \mathbb{P}_{s+a}^\infty - \mathbb{P}_{s,a}\|_{\mathrm{TV}}, \tag{16}$$

where $\mathbb{P}_{-\infty}^s$ is the joint distribution of the states $\{X_t\}_{t=-\infty}^s$ and $\mathbb{P}_{s+a}^\infty$ of $\{X_t\}_{t=s+a}^\infty$. With $\mathbb{P}_{s,a}$ we denote the joint distribution of the objects around the tensor sign, here $(\{X_t\}_{t=-\infty}^s, \{X_t\}_{t=s+a}^\infty)$ and use $\|\cdot\|_{\mathrm{TV}}$ for total variation. The process is $\beta$-mixing if $\beta(a) \to 0$ for $a \to \infty$.

MMD-mixing is closely related with equation 16 and yields lower bounds.

**Lemma 3** *A $\beta$-mixing process is MMD-mixing for any bounded kernel.*

**Proof:** This property follows by considering Hilbert space embeddings of probability distributions. In particular, Sriperumbudur et al. (2010, Theorem 21 (iii)) shows that

$$\|\mathbb{P} - \mathbb{Q}\|_{\text{MMD}} \leq C\|\mathbb{P} - \mathbb{Q}\|_{\text{TV}}, \tag{17}$$

where $C$ is the supremum of the corresponding kernel. □

The other direction does not always hold. For instance, deterministic dynamical systems are, in general, not $\beta$-mixing (Hang et al., 2017).

**Lemma 4** *A $\mathcal{C}$-mixing process with respect to the underlying function space $\mathcal{C}$ is MMD-mixing with respect to the kernel $k$, if $\mathcal{H} \subset \mathcal{C}$, where $\mathcal{H}$ is the corresponding RKHS.*

**Proof:** Following Definition 2 in Hang et al. (2018), $\mathcal{C}$-mixing is essentially defined as equation 2, where $\mathcal{F}$ is chosen as the function space $\mathcal{C}$ and $\mathcal{G}$ as $L^1$ on the natural filtration of the system. By considering smaller spaces for $\mathcal{F}$ and $\mathcal{G}$, such as $\mathcal{H}$, we directly obtain the result. □

In particular, if $k$ is the squared exponential kernel then the corresponding RKHS is well-investigated Steinwart & Christmann (2008). In particular, the RKHS is contained in common choices for $\mathcal{C}$, such as BV$(S)$, Lip$(S)$, and $C^1(S)$.

Further, recent results show that convergence in MMD metrizes weak convergence in the space of probability distributions (Simon-Gabriel et al., 2020). Thus, convergence in MMD is applicable to discrete data and Dirac distributions. This may be particularly relevant when considering mixing properties of deterministic dynamical systems even further.

In practice, mixing is usually exponentially fast in the gap $a$. In all of our numerical experiments, it was sufficient to estimate a single time shift $a^*$ in the MMD-mixing sense between two data points. The joint dHSIC estimation yields stronger theoretical properties, however, might also induce some conservatism into the estimation.

# 6 Two-sample Test for Dynamical Systems

Next, we utilize mixing to state our main result: a kernel two-sample test for dynamical systems. Due to MMD-mixing, we are able to enforce arbitrarily small dependencies between consecutive samples. In particular, we use our notion of approximately $\epsilon$-independent data (cf. equation 15) to adjust the test threshold accordingly. For $a \to \infty$, we actually recover the i.i.d. setting from Gretton et al. (2012a).

**Proposition 2** *Assume $\{X_k\}$ and $\{Y_k\}$ are stationary and MMD-mixing dynamical systems with distributions $\mathbb{P}_X, \mathbb{P}_Y$. Further, assume data $X_{a^*}, X_{2a^*}, \ldots, X_{na^*}$ and $Y_{a^*}, Y_{2a^*}, \ldots, Y_{na^*}$ are sampled i.i.d. from $\mathbb{P}_X$ and $\mathbb{P}_Y$, respectively. If we obtain for the empirical estimate equation 6 that*

$$\text{MMD}_b^2[X, Y] > \kappa(n, \alpha), \tag{18}$$

*then we can conclude with probability $1 - \alpha$ that $\mathbb{P}_X \neq \mathbb{P}_Y$. The choice of the threshold $\kappa(n, \alpha)$ is discussed extensively in Gretton et al. (2012a) and also above.*

In practice, the autocorrelations will always be greater than zero. Also, it is important that either both systems have the same mixing speed or $a^*$ is chosen with respect to the system with the slower mixing rate.

We state the main result that, in contrast to prior work, foregoes the need for independence assumptions.

**Theorem 3** *Assume the same setting as above, however, instead of i.i.d. data, we assume that $a^*$ is a time shift that yields approximately $\epsilon$-independent (cf. 15) data $X = X_{a^*}, X_{2a^*}, \ldots, X_{na^*}$ and $Y = Y_{a^*}, Y_{2a^*}, \ldots, Y_{na^*}$.*

*If we obtain for the empirical estimate equation 6 that*

$$\text{MMD}_b^2[X, Y] > \kappa(n, \alpha), \tag{19}$$

*then we can conclude with probability $1 - \alpha'$ that $\mathbb{P}_X \neq \mathbb{P}_Y$, where $\alpha' = \frac{1}{3}(\alpha + 2\epsilon)$*

**Proof:** First, we will decompose the test statistic into the i.i.d. problem and a second term that captures the dependency in the data. Assume $\bar{X}, \bar{Y}$ are i.i.d. data sets (ghost samples) that are drawn from $\mathbb{P}_X$ and $\mathbb{P}_Y$, respectively. A similar argument is frequently used for symmetrization and referred to in Gretton et al. (2012a, P. 736).

$$|\text{MMD}(\mathbb{P}_X, \mathbb{P}_Y) - \text{MMD}_b(X, Y)| \tag{20}$$

$$= |\text{MMD}(\mathbb{P}_X, \mathbb{P}_Y) - \text{MMD}_b(\bar{X}, \bar{Y}) + \text{MMD}_b(\bar{X}, \bar{Y}) - \text{MMD}_b(X, Y)| \tag{21}$$

$$\leq |\text{MMD}(\mathbb{P}_X, \mathbb{P}_Y) - \text{MMD}_b(\bar{X}, \bar{Y})| + |\text{MMD}_b(\bar{X}, \bar{Y}) - \text{MMD}_b(X, Y)| \tag{22}$$

$$= \|\text{MMD}(\mathbb{P}_X, \mathbb{P}_Y) - \text{MMD}_b(\bar{X}, \bar{Y})\| + |\|\hat{\mu}_{\bar{X}} - \hat{\mu}_{\bar{Y}}\|_{\mathcal{H}} - \|\hat{\mu}_X - \hat{\mu}_Y\|_{\mathcal{H}}| \tag{23}$$

$$\leq |\text{MMD}(\mathbb{P}_X, \mathbb{P}_Y) - \text{MMD}_b(\bar{X}, \bar{Y})| + \|\hat{\mu}_{\bar{X}} - \hat{\mu}_{\bar{Y}} - \hat{\mu}_X + \hat{\mu}_Y\|_{\mathcal{H}} \tag{24}$$

$$\leq |\text{MMD}(\mathbb{P}_X, \mathbb{P}_Y) - \text{MMD}_b(\bar{X}, \bar{Y})| + \text{MMD}_b(\bar{X}, X) + \text{MMD}_b(\bar{Y}, Y) \tag{25}$$

We use the identity $\text{MMD}_b(X, Y) = \|\hat{\mu}_X - \hat{\mu}_Y)\|_{\mathcal{H}}$ (Muandet et al., 2017, Eq. 3.31) and apply the inverse triangle inequality. The first term follows directly from Gretton et al. (2012a) (cf. eq. 7) and can be bounded by $\kappa$. By design, the time shift $a^*$ was chosen to induce the concentration

$$\mathbb{P}\left[\text{MMD}_b(X, \bar{X}) > \kappa\right] \leq \epsilon \tag{26}$$

and respectively also for $\text{MMD}_b(Y, \bar{Y})$. Thus, we obtain in total

$$\mathbb{P}\left[|\text{MMD}(\mathbb{P}_X, \mathbb{P}_Y) - \text{MMD}_b(X, Y)| > \kappa\right] \tag{27}$$

$$\leq \mathbb{P}\left[|\text{MMD}(\mathbb{P}_X, \mathbb{P}_Y) - \text{MMD}_b(\bar{X}, \bar{Y})| + \text{MMD}_b(\bar{X}, X) + \text{MMD}_b(\bar{Y}, Y) > \kappa\right] \tag{28}$$

$$\leq \tfrac{1}{3}\mathbb{P}\left[|\text{MMD}(\mathbb{P}_X, \mathbb{P}_Y) - \text{MMD}_b(\bar{X}, \bar{Y})| > \kappa\right] + \tfrac{1}{3}\mathbb{P}[\text{MMD}_b(\bar{X}, X) > \kappa] + \tfrac{1}{3}\mathbb{P}[\text{MMD}_b(\bar{Y}, Y) > \kappa] \tag{29}$$

$$\leq \frac{1}{3}(\alpha + 2\epsilon). \tag{30}$$

$\square$

**Remark 1** *By adapting the concentration results inside the kernel two-sample test, i.e., McDiarmid's inequality, we can directly embed significant autocorrelations in the test statistic and potentially be more data-efficient and have tighter bounds. These results, however, would require further technical assumptions (e.g., Assumption 3.1. in (Chérief-Abdellatif & Alquier, 2022), which is used in a different context and does not estimate mixing). Here, we focus on introducing an efficient, sound, and practically relevant statistical test for dynamical systems.*

In practice, we usually do not have access to the full state $X_k$. Instead we receive measurements $X_k' = g(X_k) + \xi_k$, where $g$ is an observation function and $\xi_k \overset{\text{iid}}{\sim} \mathbb{P}_\xi$ measurement noise. Intuitively, the function $g$ could be regarded as sensors that measure some quantity that depends on the underlying system. Thus, we could also infer different systems when, e.g., the measurement noise or the sensors are different. Further, it is not always possible to reconstruct the state, and appropriate observability assumptions would be required for this. However, this is not due to our test but an issue of the problem itself since the true underlying state is unknown. Nonetheless, we are able to apply the proposed test to measurements $X_k'$ by considering the pushforward of the measure $g(\mathbb{P}_X)$ together with $\mathbb{P}_\xi$ and correspondingly, $Y_k' = h(Y_k) + \nu_k$ with $\nu_k \overset{\text{iid}}{\sim} \mathbb{P}_\nu$ and observation function $g$.

**Proposition 3** *Assume the same setting as in theorem 2, however, with noisy measurements $X_k' = g(X_k) + \xi_k$ and $Y_k' = h(Y_k) + \nu_k$ and independent noise. If $\text{MMD}_b^2[X', Y'] > \kappa(n, \alpha)$, then we conclude that $\mathbb{P}_{X'} \neq \mathbb{P}_{Y'}$ with high probability.*

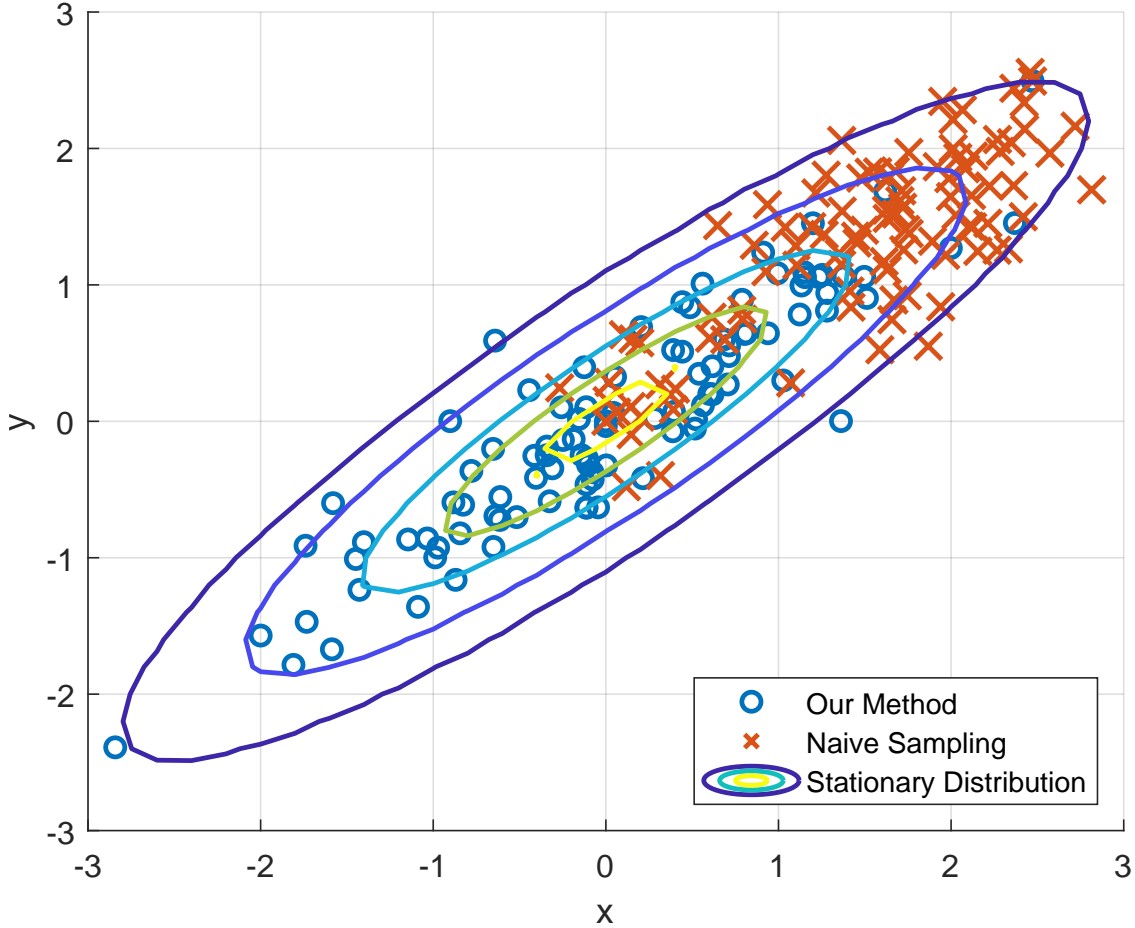

Figure 1: Scatter plot of an illustrative 2-dimensional LTI system. The system was designed to yield slow mixing times and initialized at $X_0 = 0$. The red crosses represent the first 100 states $X_0, X_1, \ldots, X_{100}$. The blue circles represent states with an enforced time shift of $a^* = 75$ between samples. The stationary distribution of the system is illustrated as a contour plot.

In general, it is not clear what states to choose for an appropriate representation of a dynamical system, e.g., to accurately model human walking. If we obtain rich information through sensor measurements then this can often be sufficient for subsequent downstream tasks (cf. section 8).

## 7 Illustrative Examples

In this section, we illustrate two critical properties of our method: i) respecting the estimated time shift $a^*$ yields samples whose distribution is indistinguishable from the stationary distribution, ii) violating the estimated time shift $a^*$ leads to clustering effects that skew and bias the empirical distributions. More details on all experiments (deterministic and stochastic systems) are provided in the appendix.

In practice, it is usually sufficient to consider data from two points in time $\mathbb{P}_s$ and $\mathbb{P}_{s+a}$—in particular, when kernel two-sample testing is also based on points and not subtrajectories. Thus, we estimate $a^*$ based on MMD-mixing (cf. def. 7) and (unless stated otherwise) pick the first time shift that is below the test threshold. Further, following Gretton et al. (2012a), we consider squared exponential kernels and chose the bandwidth based on the well-established median heuristic. Hyperparameter optimization can further improve the power of the test. We excluded such a discussion since this is not the main emphasize of our work

and orthogonal to our contributions – how can kernel two-sample tests be adapted to dynamical systems and how can a suitable notion of mixing be estimated from data.

## 7.1 Linear Time-invariant System

Linear time-invariant (LTI) systems are prevalent in control and systems theory due to many analytically tractable properties. In particular, we can explicitly determine the stationary distribution (cf. equation 39 in the appendix) and, thus, draw i.i.d. samples. Consider the dynamics

$$X_{k+1} = AX_k + \epsilon_k, \tag{31}$$

where $\epsilon_k \overset{\text{iid}}{\sim} \mathcal{N}(0, \Sigma)$. Further, assume all eigenvalues of $A \in \mathbb{R}^{d \times d}$ are located within the unit circle and $x_0 = 0$ to avoid potential transient behavior. We can now quantify the speed of mixing directly through the eigenvalues of $A$ and $\Sigma$. If $A$ has eigenvalues close to the boundary of the unit sphere, then this results in slow mixing. The same holds for small process noise. On the contrary, small eigenvalues of $A$ and large noise result in rapid mixing. An intuitive corner case is $A = 0$, which yields perfectly independent samples.

In fig. 1, we illustrate the behavior of a two-dimensional slowly mixing system (31). In red, we plot the first 100 states of the system $X_1, X_2, \ldots, X_{100}$, and in blue, states with an enforced time shift of $a^* = 75$ between consecutive samples $X_{a^*}, X_{2a^*}, \ldots, X_{100a^*}$. The estimated time shift $a^* = 75$ is obtained in the MMD-mixing sense as described in sec. 5.2 and ensures that the HSIC is below the test threshold (cf. fig. 4 in the appendix).

In fig. 1, we further show a contour plot of the stationary distribution. Samples that were drawn based on our method coincide with the stationary distribution. The first 100 states, on the other hand, cluster in one region of the state space and are subject to heavy auto-correlations. The red crosses are clearly not representative of the stationary distribution. To investigate this further, we applied kernel two-sample tests to distinguish samples that are directly drawn from the stationary distribution and samples drawn based on our method with appropriate time shifts. As expected, this turned out to be impossible, and we cannot distinguish between the two data sets. Details are given in the appendix.

We want to emphasize that mixing can be arbitrarily slow. In particular, it is possible that the system does not mix at all (Simchowitz et al., 2018). With the proposed method, we would notice this since we would not be able to estimate an appropriate $a^*$ due to substantial remaining correlations in the data. Thus, we can decide whether the kernel two-sample test is applicable or not. We have also constructed non-mixing examples and obtained a constant HSIC that does not decrease over time (cf. appendix).

## 7.2 Lorenz Attractor

To illustrate the usefulness of the new mixing notion we present the example of the Lorenz system, which is illustrated in fig. 2 and given by the following equations:

$$\dot{x} = 10(y - x) \tag{32}$$

$$\dot{y} = 28x - y - xz \tag{33}$$

$$\dot{z} = xy - \frac{8}{3}z. \tag{34}$$

The Lorenz attractor is a famous chaotic and deterministic dynamical system that is known to mix in a topological sense (Luzzatto et al., 2005). Other notions, such as $\beta$-mixing, are too strong and not suitable here. In the appendix, we provide empirical evidence that the Lorenz system mixes with respect to the herein introduced notion of MMD-mixing. Connecting topological mixing on a rigorous level with MMD-mixing remains for future work.

Further, we show numerically that we can distinguish between two systems with slightly different parameters and obtain the required properties of the kernel two-sample test. For the estimated $a^*$, the test is well-behaved. When we chose the estimated time shift too small, then the amount of false positives explodes.

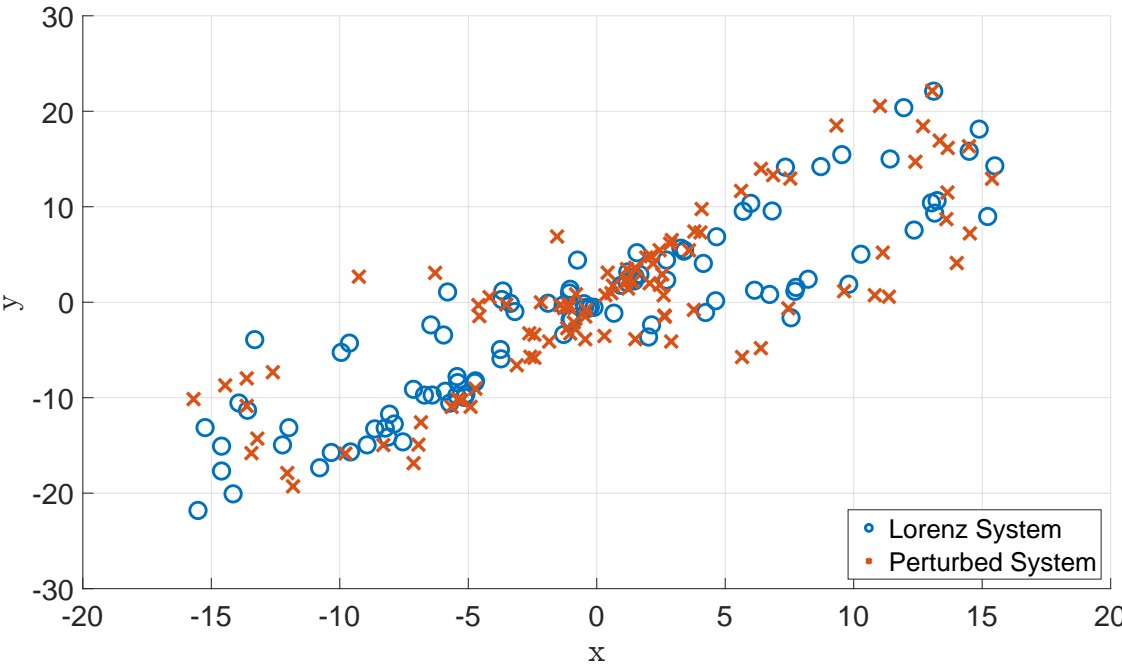

Figure 2: Scatter plot of the classical Lorenz system in blue circles (eq. equation 32—equation 34) and samples from a system with slightly perturbed parameters in red crosses. Both systems are randomly initilized and sampled at time instances $t_1 = 20, t_2 = 40, \ldots, t_{100} = 2000$. To the human eye, the distributions look slightly different.

### 7.3 $\mathcal{C}$-mixing Systems

We also consider the three examples that are discussed in Hang et al. (2018) and are provably $\mathcal{C}$-mixing. Due to the Gaussian kernel that we use and the choices for $\mathcal{C}$ (Lip($S$) and BV($S$), cf. Hang et al. (2018) for details), $\mathcal{C}$-mixing directly implies MMD-mixing. The empirical results confirm the MMD-mixing property.

We considered the following systems:

$\beta$-map: For $\beta > 1$ and $x_0 \in (0,1)$, the dynamical system is defined by

$$x_{k+1} = \beta x_k \mod 1. \tag{35}$$

**Logistic map:** For $x_0 \in (0,1)$, the logistic map is defined by

$$x_{k+1} = 4x_k(1 - x_k). \tag{36}$$

**Gauss map:** For $x_0 \in (0,1)$, the Gauss map is defined by

$$x_{k+1} = \frac{1}{x_k} \mod 1. \tag{37}$$

For all examples, the speed of mixing is extremely fast and after $a^* = 10$, the data is close to independent. We initialized $x_0$ uniformly on the interval $(0,1)$ and used $\beta = e$.

Interestingly, the Lebesgue densities of the stationary distributions are also known and stated in Hang et al. (2018). This would allow for kernel two-sample testing, exactly as done for the OU-process in Sec. A.1.1. Since the mixing is extremely fast here, we expect the same result—time shifted data that respects the speed of mixing is indistinguishable from data that has been drawn directly from the stationary distribution.

# 8 Experimental Example—Human Walking

We apply the developed kernel two-sample test to real-world experimental data [1]. We consider gait data of human subjects walking on a treadmill. Detecting characteristics and alterations in human gait is a highly relevant problem in disease prediction, diagnosis and progress monitoring as well as in biometrics (Nguyen et al., 2019; Gaßner et al., 2020; Muro-De-La-Herran et al., 2014). An example data set with a known ground truth label is obtained by letting subjects walk with and without a knee orthosis. Our goal is to classify each measured trajectory correctly with the labels *orthosis* and *no orthosis*.

## 8.1 Data Collection

The inertial measurement unit (IMU) data of foot motion were collected from 38 healthy subjects without any restrictions in gait or illnesses that affect their walking ability. The data collection was conducted in the motion analysis laboratory of one of the authors' universities on a Mercury Med treadmill. The IMU sensors were attached to the test subjects' shoes using velcro straps. The measurements were taken for 90 seconds each trial under the following conditions: walking at very slow ($1.5\,\mathrm{km\,h^{-1}}$), slow ($3\,\mathrm{km\,h^{-1}}$), slow with simulated gait pathology, and normal walking speed ($5\,\mathrm{km\,h^{-1}}$). For the simulated gait pathology, the mobility of the left knee joint was restricted using a knee orthosis, which was fixed in a neutral position to disable further extension or flexion of the joint. The subjects were asked to stand still with both feet next to each other for 3 seconds at the beginning and the end of each trial, Before the trials, the subjects were able to practice walking on the treadmill. They were allowed to use the handrail of the treadmill if necessary. For three subjects, the orthosis experiment could not be carried out. An approval from the local ethics committee was obtained.

## 8.2 Description of the Statistical Test

We consider the raw gyroscopic data of the left foot for 35 subjects. The gyroscopic data is three-dimensional and consists of roughly $14\,000$ data points per trajectory.

### 8.2.1 Mixing Properties

First, we quantify the mixing properties of human walking. We estimate MMD-mixing (cf. sec. 5.2) by applying the HSIC to the 35 subjects. We consider the trials with and without the orthosis simultaneously, which yields 70 independent trajectories. We draw an initial point $X_k$ from a uniform distribution between $k = 2000$ and $k = 4000$ and fix that point for all trajectories. Afterward, we compute $\mathrm{HSIC}(X_k, X_{k+a})$ for various values of $a$. In fig. 3, the results are illustrated and we can see the decrease of dependencies. To exclude numerical artifacts, we repeat the estimation of the mixing properties 50 times with randomly chosen initial points.

### 8.2.2 Classification

We compare MMD-based classification against standard baselines for the 70 trajectories.

**MMD-based classification:** We choose one trajectory of interest, for which we forget the correct label, and separate it from all other trajectories, for which the correct label is known. We pick a random initial point $X_0$ uniformly distributed between $k = 2000$ and $k = 3000$. After time shifting the data with respect to $a^*$, we estimate the MMD equation 6 between the trajectory of interest and all other trajectories. Then, we use the label of the trajectory with the smallest MMD to label the unlabeled trajectory. Intuitively, unrestricted trajectories look more similar among themselves than trajectories with a restricted knee, and vice versa.

**Baseline:** We compare the proposed approach to common baselines for classification of biomedical data (Bidabadi et al., 2019; Misgeld et al., 2015; Tien et al., 2010). We consider the following features:

- Maximum and minimum value of each dimension;

---

[1]The data will be made available upon acceptance

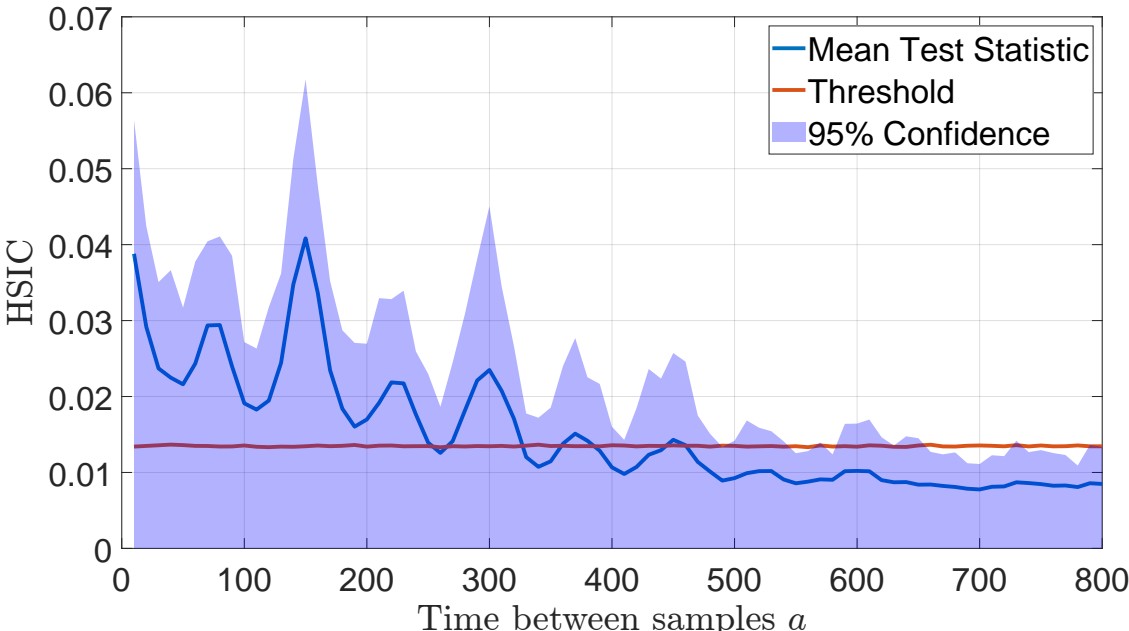

Figure 3: Mixing properties of gait data. On the $x$-axis, we depict the time shift $a$ between consecutive samples. One time step corresponds to 0.01 seconds. The $y$-axis shows the dependence between data points with respect to the corresponding time shift. The initial point is randomized, and the estimation is repeated 50 times. Depicted is the mean of the test statistic and the 95% upper confidence bound. We also show the threshold $\kappa$ of the independence test. When the blue line is below the red line, it is not possible to infer statistical dependence between the data points.

- The 4 largest frequencies based on a Fourier transform;
- The 2-norm over time and the state dimensions.

In total, this results in 19 features for each trajectory that are used to train linear classifiers—support vector machines (SVM) and logistic regression (LR). Further, we use a 3-fold cross-validation technique. We repeat the training also 1000 times and report the average accuracy and standard deviation in Table 1.

### 8.3 Results

Our empirical analysis reveals that human walking mixes with respect to MMD-mixing (cf. sec. 5.2). Further, as illustrated in fig. 3, we can effectively estimate the speed of mixing. After roughly five footsteps, the dependence of data to its past is mostly gone, and we can treat data as independent.

For MMD-based classification, we use a time shift of $a^* = 400$. This results in 25 points per trajectory. A larger choice of $a^*$ around 600 would be closer to our theoretical results. However, due to the limited amount of data, this would reduce the number of available samples even further.

We run all classification algorithms 1000 times and report the average accuracy and standard deviation in Table 1. Our method achieves the best accuracy, and we are able to classify 99.99% of the subjects with *no orthosis* correctly. Some very few subjects are repeatedly misclassified when walking with the orthosis, which might be explained using futher insights and data analysis. For the other methods, in contrast, there is no apparent structure in the errors.

### 8.4 Discussion

The above results show that the proposed method works well on a practically relevant non-trivial problem and beats reasonable baselines. We spent a reasonable amount of time on designing good features in the

Table 1: Classification accuracy for labeling the trajectories correctly into the labels *orthosis* and *no orthosis*. Mean accuracy with standard deviation over 1000 repetitions.

| Our method | SVM | LR |
|---|---|---|
| **95.7**% $\pm$ 2.4% | 86.9% $\pm$ 4.4% | 92.5% $\pm$ 3.2% |

comparison. While the accuracy of the linear classifiers could potentially be improved by adding additional features, designing such features requires more insight into the problem and system properties, which is unavailable in many applications. In order to improve the accuracy of our method, it would suffice to add more data (i.e., consider longer trajectories). Further, it can directly be applied to a range of similar problems.

Classification and clustering algorithms based on the MMD can be applied in more general settings (Jegelka et al., 2009). Thus, our proposed nearest-neighbor approach for dynamical systems should generalize to more sophisticated clustering algorithms, which could yield unprecedented insights into the behavior of complex dynamical systems.

## 9 Conclusion

We propose a kernel two-sample test for dynamical systems with deep connections to a new type of mixing in MMD. The proposed method is straightforward to use, has only a few parameters, and is model-free. In particular, we are able to estimate the speed of mixing from data in a relevant norm, which was previously not possible. The method is tailored to dynamical systems for which we have access to multiple independent and long trajectories. The flexibility and relevance of the proposed method are demonstrated numerically and experimentally on raw motion sensor data. The presented results show the potential for biomedical and engineering applications, which we plan to explore in future work.

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

# A  Appendix

## A.1  LTI Systems

There are several aspects that we kept short in the main paper and address in the following. We consider the dynamics

$$X_{k+1} = AX_k + \epsilon_k, \tag{38}$$

where $\epsilon_k \overset{\text{iid}}{\sim} \mathcal{N}(0, \Sigma)$. Further, assume all eigenvalues of $A \in \mathbb{R}^{d \times d}$ are located within the unit circle.

**Stationary Distribution:** The stationary distribution of an LTI system is Gaussian with expected value zero. The Gaussian distribution follows from the Gaussian noise and linear structure of the system. The expected value can be computed by leveraging that all eigenvalues of $A$ are located within the unit circle. Obtaining the variance is more involved. It can be expressed as the solution to the following Lyapunov equation in $Z$ (Schluter et al., 2020, Equation 7):

$$AZA^\intercal - Z + \Sigma = 0, \tag{39}$$

where $A$ is the system matrix and $\Sigma$ the covariance matrix of the process noise.

### A.1.1  Comparison to Stationary

We investigate if we can, based on a kernel two-sample test, distinguish between time shifted samples with respect to $a^*$ and i.i.d. samples from the stationary distribution.

**Setup:** We create 500 randomly generated LTI systems with a random dimensionality between 1 and 100. For each system we create $m = 250$ independent trajectories and sample $n = 20\,000$ points for each trajectory. All systems are initialized in $X_0 = 0$ to avoid transient effects. The decay of dependence is quantified in

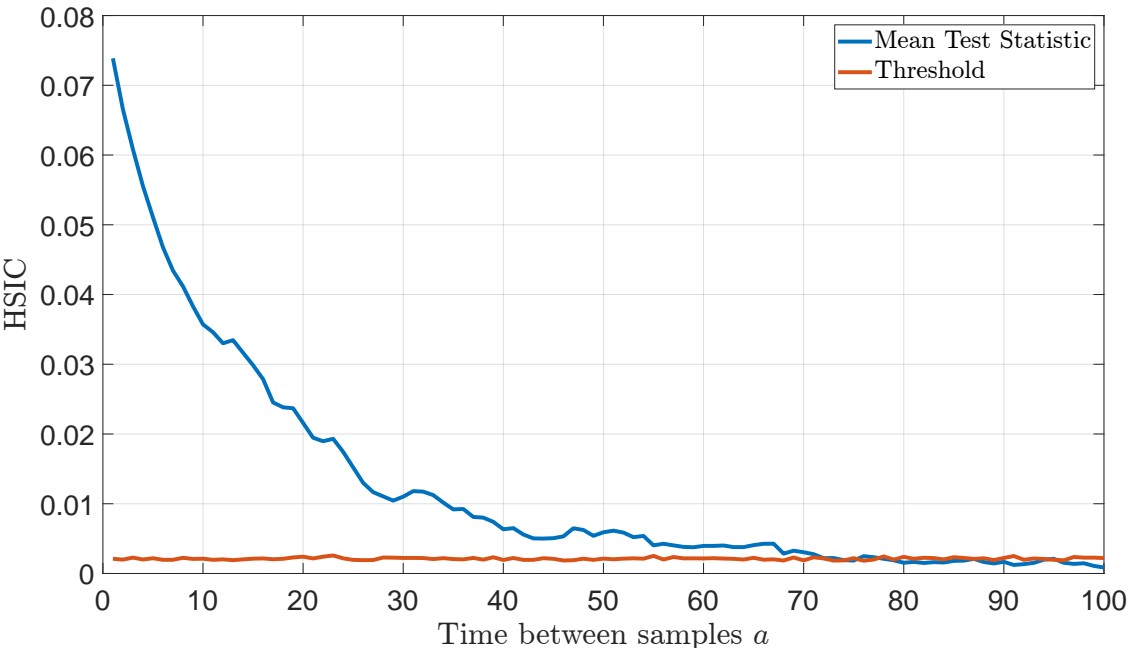

Figure 4: Mixing properties of the LTI system that is used to create fig. 1 in the main paper. On the $x$-axis, we depict the time shift $a$ between consecutive samples. The $y$-axis shows the dependence between data points with respect to the corresponding time shift. At $a^* = 75$ the test statistic is below the threshold.

the MMD-sense for one gap (cf. sec. 5 of main paper). We use data from the end of the trajectory to avoid numerical artifacts due to the identical initial values.

Next, we describe how we generate the system matrices.

**Sampling $\Sigma$:** The entries for the covariance matrix are drawn from a standard multivariate normal distribution. Since the matrix is supposed to yield a covariance matrix, we require symmetry and positive definiteness. Thus, we denote $\Sigma'$ as the matrix drawn from the normal distribution and define $\Sigma = 0.5(\Sigma' + \Sigma'^\intercal)^2$. To control the magnitude of noise, we scale the matrix with the largest eigenvalue of $\Sigma$.

**Sampling $A$:** The system matrix $A$ is required to have eigenvalues within the unit sphere. To achieve this, we draw the entries of $A$ from a uniform distribution and extend the system with a control input

$$X_{k+1} = AX_k + Bu_k + \epsilon_k. \tag{40}$$

The control matrix $B$ is set to the identity matrix and the control input as a standard linear quadratic feedback controller $u_k = -Kx_k$. This yields the closed loop dynamics

$$X_{k+1} = (A - BK)X_k + \epsilon_k. \tag{41}$$

The feedback gain $K$ can be computed to minimize a linear quadratic cost function. By adjusting the weights of the cost function, we can indirectly adjust the eigenvalues of the closed loop system matrix $(A - BK)$. We set the weight matrix for the state cost $Q$ to the identity matrix and the control cost to $R = 10^7$. This makes it very expensive to apply large control inputs and magnitude of the eigenvalues of $(A - BK)$ stays close to 1. This implies slow mixing and further, by considering $R \to \infty$, we can make this arbitrarily slow.

**Results:** First, we use the $m = 250$ trajectories to estimate the mixing speed $a^*$. We choose $a^*$ as the first time instance at which the test statistic is below the test threshold.

For the kernel two-sample test, we draw 100 points from the first trajectory that respect the time shift $a^*$. We also draw 100 points directly from the stationary distribution ($\mathcal{N}(0, Z)$, cf. equation 39).

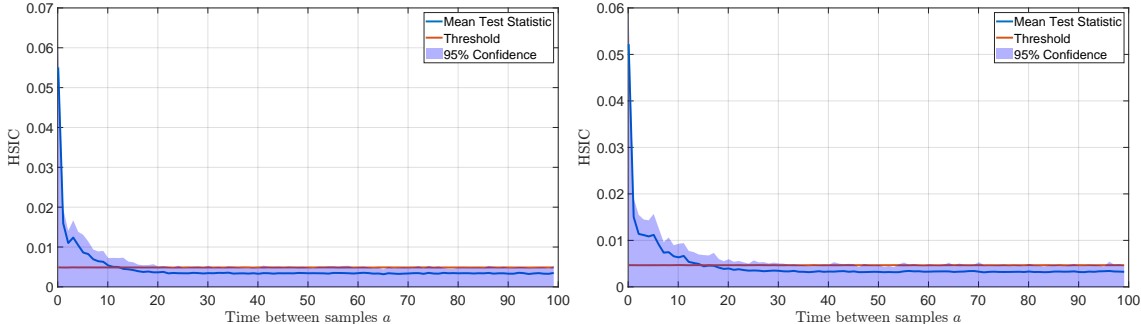

Figure 5: Mixing properties of the Lorenz system. Left plot with parameters as in equation 32—equation 34 and on the right, we adapted the parameter in equation 32 to 6. On the $x$-axis, we depict the time shift $a$ between consecutive samples. The $y$-axis shows the dependence between data points with respect to the corresponding time shift. The initial point is randomized, and the estimation is repeated 100 times. Depicted is the mean of the test statistic and the 95% upper confidence bound. We also show the threshold $\kappa$ of the independence test. When the blue line is below the red line, it is not possible to infer statistical dependence between the data points.

From the 500 systems we considered overall, we only obtained 4 false positives, which shows the high precision of our proposed test. Due to the probabilistic nature of these experiments, we could obtain systems with arbitrarily slow mixing times and, subsequently, very long $a^*$. Thus, we decided to fix a maximum $a^*$ as $a_{\max} = 200$, and ignore all systems with larger $a^*$. We obtained 81 systems that mix too slowly, i.e., $a^* > a_{\max}$.

### A.1.2 Details for fig. 1 in the main paper:

In fig. 4, we show the mixing properties of the system that yields $a^* = 75$. We used the same setup as in sec. A.1.1 with some modifications. We chose $R = 10^{10}$ and divided $\Sigma$ by $10\lambda_{\max}^2$, where $\lambda_{\max}$ is the largest eigenvalue of $\Sigma$. Further, to be able to better visualize the samples and the stationary distributions, we fixed the dimension to two.

The randomly generated system matrices are

$$A = \begin{pmatrix} 0.2345 & 0.8609 \\ 0.7298 & 0.1316 \end{pmatrix}, \Sigma = \begin{pmatrix} 0.0378 & 0.0135 \\ 0.0135 & 0.0971 \end{pmatrix}. \tag{42}$$

### A.2 Lorenz System

To perform kernel two-sample testing, we slightly change the parameters in the Lorenz system by decreasing the coefficient in equation 32 from 10 to 6 to obtain a second slightly different system. The mixing analysis is done for both systems. The attractors of both systems look optically very similar. The attractor can be interpreted as the stationary probability distribution of the state in some sense.

### A.2.1 Mixing properties

We estimate the mixing properties of the Lorenz system in the MMD-mixing sense for one time shift $a$ (cf. Sec.5).

**Initial points:** We sample from an uniform distribution $\mathcal{U}([-0.5, 0.5] \times [-0.5, 0.5] \times [20, 21])$ to initialize the starting point $X_0$.

**Data:** We use a standard ODE solver[2] to obtain a solution to the Lorenz system. Due to variable step sizes within the solver, we interpolate the solution to obtain samples with a fixed discretization in time. We consider the time horizon $t \in [0, 200]$ and create 2001 samples (with a fixed time step of 0.1).

---

[2]ode45 in Matlab

**Repetitions:** We create $M = 100$ independent trajectories to estimate the mixing properties. The experiment is repeated $N = 100$ times to investigate deviations in the decay of the dependence.

**Estimating mixing:** To avoid numerical artifacts due to the initial points and potential transients, we consider data from the end of the trajectory. Thus, we sample at $t_{\mathrm{end}} = 200$ and at $t - a$ for various values of $a = 0.1, 1.1, 2.1, \ldots, 99.1$ with respect to the continuous time index $t$.

**Results:** We depict the decay of dependence in fig. 5. After waiting for $a^* = 20$, the dependence in the data is not detectable anymore. Since the decay is not necessarily monotonic, we consider significantly higher time shifts up to $a = 99.1$. The dependence does not increase again, which indicates that the system is mostly mixing in the MMD-sense. Of course, this does not prove that the Lorenz system mixes and it remains to be shown rigorously. Nonetheless, these results are promising and provide empirical evidence.

### A.2.2 Kernel Two-sample Test

We try to distinguish between the Lorenz system given in equation 32—equation 34 and a slightly disturbed system where we change the parameter in equation 32 from 10 to 6. Based on the previous mixing analysis (cf. fig. 5) we set the time shift $a^* = 20$. This yields approximately independent samples for both systems.

We create two trajectories of length $t_{\max}$ and pick $n$ points that respect the time shift $a^*$ as illustrated in fig. 2. We repeat all experiments 100 times. We start the sampling after $t = 20$, which gives the system enough time to converge to the stationary distribution.

**Accuracy:** We use $t_{\max} = 6000$ and pick $n = 300$ points from both system. We achieve 95% accuracy in detecting different systems.

**False positives:** We consider two trajectories that were generated by the classical Lorenz system (equation 32—equation 34. The initial points for both trajectories were random and different. This setup yields 2.67% false positives, which is less than the $\alpha$-level of 5% that we used.

Next, we investigate what happens if we violate $a^*$. We choose $n = 100$ and $t_{\max} = 30$. Thus, we sample 100 points in the time interval $t \in [20, 30]$. This clearly violates the estimated $a^*$ and indeed, we obtain 51% false positives. Essentially, this makes the test useless when $a^*$ is severely violated and thus, we want to emphasize again that it is critical to estimate $a^*$. Further, through an appropriate choice of $a^*$ we inherit all the rich theoretical properties of kernel two-sample testing.

### A.3 Non-mixing System

We construct a system that does not mix in the MMD sense and is also not expected to mix. However, the system is well known to be ergodic and stationary. In particular, we consider a dynamical system that moves on a circle with a radius of one and steps of length $\frac{\pi}{10}$. We create $m = 100$ randomly initialized points $\theta_0$ and iterate them for $n = 100$ timesteps with the dynamics following

$$\theta_{k+1} = \theta_k + \frac{\pi}{10}, \tag{43}$$

and

$$X_{k+1} = \begin{pmatrix} \cos(\theta_k) \\ \sin(\theta_k) \end{pmatrix} \tag{44}$$

We show the mixing properties in fig. 6. The dependence between data points stays constant and does not decrease and we detect this. Thus, we correctly identify systems that are not mixing in the MMD sense.

We have also tried different increments instead of $\frac{\pi}{10}$, such as $\frac{e}{10}$ and also $\frac{1}{10}$, which all resulted in the same outcome.

### A.4 Implementations

Since our method is leveraging results from standard kernel two-sample testing and the HSIC, we directly used existing implementations without modifying them.

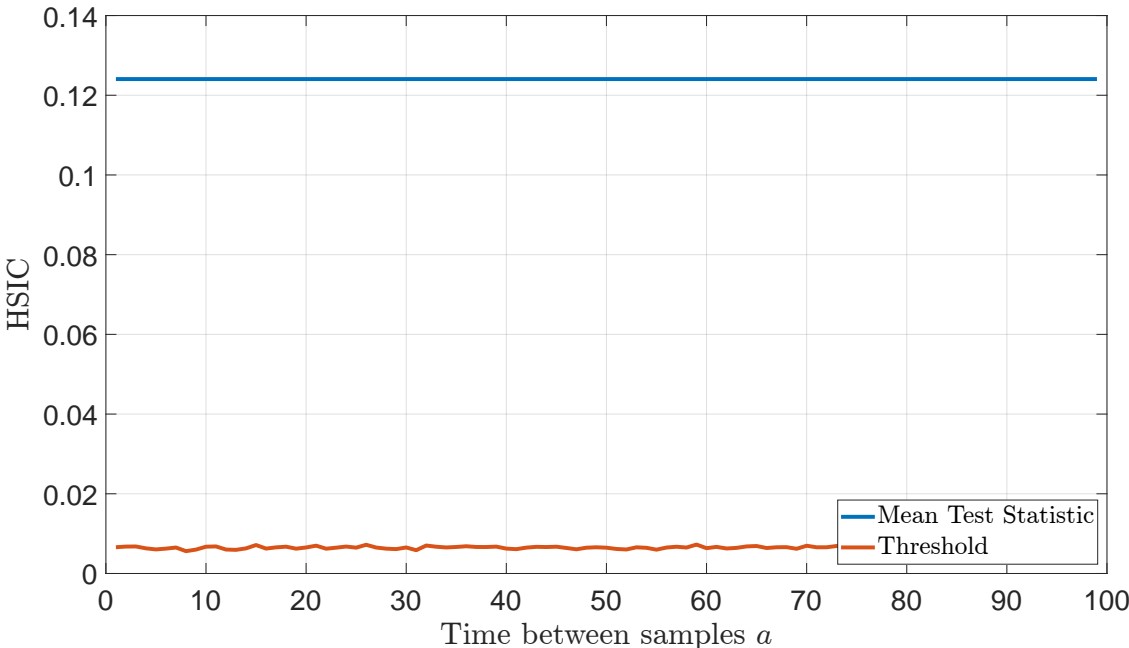

Figure 6: Mixing properties of a dynamical system that moves on a circle. The dependency between data points does not decrease and stays above the threshold.

**Kernel Two-sample Test Implementation:** We used the Matlab implementation: `http://www.gatsby.ucl.ac.uk/~gretton/mmd/mmd.htm` and the standard hyperparameters without any tuning. We used the significance level $\alpha = 0.05$ for all experiments.

**HSIC Implementation:** We used the Matlab implementation: `http://people.kyb.tuebingen.mpg.de/arthur/indep.htm` with standard hyperparameters and $\alpha = 0.05$ for all experiments.

