# OpenReview forum: "A Kernel Two-sample Test for Dynamical Systems"
_TMLR — Rejected by TMLR_

### Review · Reviewer_SL5a · 2022-08-26

**Summary Of Contributions:**

In this article, the authors propose a new statistical test for comparing time series with dependencies between measurements, i.e., a test that goes around the usual iid assumption about the data points of the time series. The test is based on the weaker assumption that, after a long enough time shift, the samples are independent, and amounts to running the usual MMD test on subsampled time series such that the consecutive subsamples are separated by a good time shift (which can be computed and selected using standard HSIC confidence intervals for various time shifts). Under these assumptions, the authors prove some theoretical guarantees for calibrating the test and controlling the type I error, and finally provide a few experiments showing the test efficiency.

**Broader Impact Concerns:**

I have no concerns on the ethical implications of the work.

**Requested Changes:**

I have a few comments:

---I am curious about running times. As far as I understand, one would need to run confidence intervals for HSIC for several different time shifts, which sounds expensive. How does this compare to the competitor method mentioned in Remark 1?

---Concerning time shifts that yield epsilon independent data: are there theoretical examples that provably satisfy this assumption? This would better motivate the method. As far as I understand, the examples provided in Figures 1 and 2 are only empirical. Also, is there any interpretation or intuition about the fact that the data presented in Section 8 is indeed epsilon independent for a good enough time shift?

---Being able to subsample necessitates long time series with a lot of measurements. Do the authors have a strategy for short time series with just a few data points?

---The classification results of Section 8 are quite nice. I wonder if similar scores could be obtained on general time series when almost nothing is assumed about the data (i.e., we do not know a priori if the data is iid or not). This would help making the method a standard tool of time series libraries. For instance, how does the proposed classification method compare to nearest neighbor techniques on the UCR public library (https://www.cs.ucr.edu/%7Eeamonn/time_series_data_2018/)?

**Strengths And Weaknesses:**

Even though the method is simple, the authors are able to provide nice guarantees, as well as experimental details for the calibration procedure. I think the proposed approach is a nice addition to the literature, as removing the iid assumption is quite hard, yet often encountered in real world scenarios. Moreover, the article is very clear and easy to read.

---

> ### Author Response · Authors · 2022-10-05
> **Reply to Review of Paper277 by Reviewer SL5a Part 1**
>
> We thank the reviewer for valueable suggestions.
>
> - Running times
>
> Thank you for this important technical question. The running time of the HSIC is actually quite fast. In particular, it scales in the number of independent trajectories. Further, all evaluations for different time shifts can be computed in parallel.
> The objective of running the HSIC for different time shifts is determining a large enough shift $a^*$ that yields approximately independent data.
> In practice, we do not need to compute this for all possible shifts. Finding one shift that is large enough is sufficient. In the paper, we have computed the HSIC in a high resolution (with confidence bounds) to illustrate the decrease in dependencies.
>
> The method mentioned in Remark 1 does not estimate mixing from data.
> A similar notion of mixing in RKHS is introduced for the purpose of parameter estimation, which is a different problem.
> Their technical formulation of mixing could be helpful to derive sharper bounds for our Theorem 3. However, the technical and mathematical implications are nontrivial and require careful evaluation, which we believe are out of the scope of this paper.
>
> We have reformulated the original sentence: "These results, however, would require further technical assumptions" to "These results, however, would require further technical assumptions
> (e.g., Assumption 3.1. in (Chérief-Abdellatif \& Alquier, 2022)), which is used in a different context and does
> not estimate mixing."
>
> - epsilon independent
>
> Theoretical examples: Data that is i.i.d. is clearly also $\epsilon$-independent. Following up on this, we can correlate the joint distribution of i.i.d. data to artificially construct data that is $\epsilon$-independent.
>
> We have added the following sentence to the paper to provide more intuition:
> "Independent and identically distributed data naturally satisfies the above property when $\epsilon=\alpha$ and $\kappa$ is chosen correspondingly for a level-$\alpha$ kernel two-sample test."
>
> In practice: we can never guarantee that data is independent or drawn from the same distribution. By considering the HSIC over time shifts, we can (with high probability) determine that data is too correlated to perform downstream tasks such as two sample testing.
>
> The examples in Figure 1 and 2 are indeed empirical. Deriving analytically if a system mixes or not is highly nontrivial and determining the actual speed of mixing is even harder.
> Further, these results can render useless in the real world since there is usually a substantial discrepancy between simulation models and the real world behavior of a system.
> Thus, in this paper, we present an algorithm that can estimate mixing times directly from real-world data and show that this enables reliable two-sample testing for dynamical systems.
>
> We have actually discussed the walking results with colleagues that work on bipedal locomotion and in particular, model predictive control. In that community, it is a well-known heuristic that predictions become useless after roughly three walking steps. This is usually used to limit the amount of computations and having a feasible planning horizon.
> Our results are consistent with these insights.
> After three steps, data is mixed.
> In our interpretation, the oscillatory behavior of the HSIC in Figure 3 can be matched to the steps and certain parts of the step can be predicted more reliably than others.
> Since this interpretation is somewhat speculative and requires further investigation, we have decided not to include this discussion in the paper. However, we have looked into this to verify that on an intuitive level, gait data does mix.
>
> - Short time series
>
> This is indeed (for now) a disadvantage of our method.
> We are working on extensions that transform the data into the frequency domain through Fourier transformation and compare the spectrum through similar kernel two-sample tests. Our preliminary results look promising, but the approach
> requires another set of theoretical results and is designed for a different problem class than the one considered here, which we hope to publish in future work.

---

> > ### Author Response · Authors · 2022-10-05
> > **Reply to Review of Paper277 by Reviewer SL5a Part 2**
> >
> > - UCR Dataset
> >
> > We are not claiming that we propose the single best method for time series classification and further, strongly believe that depending on the specific system properties and problem requirements, different algorithms thrive.
> > Ours is designed for problems, where we have access to many independent and long trajectories.
> > For this type of systems, our method performs very well.
> > We discuss in detail the necessary assumptions and we show what can go wrong for appropriate systems that fit in this problem class. Further, we validate our method on real-world data and show that it performs well for a suitable system.
> >
> > Thank you for pointing us at the excellent data set.
> > We have analyzed the earthquake data since it was the first sensor-based dataset that consists of two labels. Interestingly, the data, which is labeled as no major event does not mix. When considering the other set of trajectories, we obtain fast mixing behavior. Intuitively, it makes sense that active earthquakes cause strong movements and thus, mixing. For the other trajectories, there is less movement. It would be interesting to investigate the non mixing class for longer time scales.
> > Moreover, this system is not stationary since at some point an earthquake appears.
> >
> > We have applied the kernel two-sample test without subsampling, even though the system is non-mixing and non-stationary.
> > In detail, we compute the pairwise MMDs between the test set and the train set. Afterward, we pick the trajectory with the smallest value and assign the label to the trajectory from the test set.
> > Additionally, we compute the mean over all the MMD values that are used for classification. There are some trajectories that look very different from all the others and show an extremely high MMD. These are therefore excluded and moved to the label unknown.
> >
> > Overall, we classify $51,08\%$ of the trajectories correctly, $22,30\%$ as unknown, and $26,62\%$ wrong. When looking at the two classes separately, we obtain for the "no major event" label the following results: $65,38$ correct, $22,12\%$ unknown, and $12,5\%$ wrong. For the trajectories labeled as earthquakes: $8,57\%$ are correct, $22,86\%$ unknown, and $68,57\%$ are wrong.
> >
> > The weaker performance of our test when compared to a nearest neighbor method can in parts already be explained by the non-stationarity especially of the data containing earthquakes. Further, we here propose a two-sample test, i.e., a method that is supposed to decide whether two samples are drawn from the same distribution. Such a method should only be used for classification if it can reasonably be assumed that samples with the same label follow the same distribution while samples with different labels come from different distributions. For the earthquake dataset, this is not a reasonable assumption. The labels are determined by a value on the Richter scale, where everything above 4 is classified as an earthquake. However, trajectories with different values on the Richter scale, even if all of them are either above or below 4, may very well follow completely different distributions.
> >
> > Because of the problems of this data and because we do not think it would add substantial insight to the paper compared to the other results, we decided to not include the datasets. We will keep this for future investigations.  Thanks again for pointing us to the dataset.

---

### Review · Reviewer_bTrg · 2022-09-13

**Summary Of Contributions:**

This paper proposes an extension of kernel two sample tests to dynamical systems. Most comparisons of data from dynamical systems require strong assumptions and/or estimating intermediate model parameters explicitly. Here, the authors propose a new mixing criteria assumption for dynamical systems based on maximum mean discrepancy (MMD). This allows them to subsample trajectories from each system until the samples are approximately independent, and then apply standard results for statistical hypothesis testing via kernel MMD. Experimental results on synthetic examples and a proprietary human gait dataset show the efficacy of the mixing criteria, independence test, and performance on a downstream classification task.

**Broader Impact Concerns:**

No major concerns with broader impact, but the authors could comment on the stability/robustness of nearest neighbor methods

**Requested Changes:**

### Critical
- Provide more details on the kernel family and hyperparameters (e.g. kernel width) used in MMD computation. How sensitive is the approach to these modeling choices?
- Report additional classification metrics such as per-class accuracy, precision, recall, and AUC.
- Clarify where MMD is used with samples from 1 trajectory, and where it is used with $m$ trajectories from $X$ and $Y$. For example, is Section 5.1 or 5.2 used in Experiment 8.2.1? How is $n$ in the hypothesis test threshold related to the number of trajectories $m$? If necessary, write Equation (6) in terms of joint distribution and joint kernel.
- A better experimental evaluation would include strong baselines with learned feature representations, such as another kernel method or a neural network.

### Non-Critical
- Additional experiments on a standard, public dataset
- Additional experiments comparing noisy measurements to a model, following the setting of Proposition 3 and Section 1, paragraph 5
- Evaluate the the proposed method using $k$-nearest neighbor classifiers for k>1
- Discuss/comment on extending from discrete-time to continuous-time dynamical systems

### Typos
- Ppage 2: "electro- myography-based" should be "electro-myography-based"
- Figure, Theorem, Appendix, etc. should be capitalized when referenced in the text
- $h$ and $\nu$ are used in Proposition 3 without definition


**Strengths And Weaknesses:**

### Strengths
- Simple, elegant technical contribution
- Cites previous work, and explains how novel contributions are different
- Generally well-written
- Illustrative synthetic examples

### Weaknesses
- Experiments: Classification results would be improved if more details on the setup and report additional performance results. For example, MMD still requires "feature engineering" in the form of selecting a kernel family and hyperparameters. Also, the paper would benefit from additional experiments for the settings described in the introduction.
- Clarity: Some concepts have thorough background information (ergodic process, $\mathcal{C}$-mixing), while some terms are used without explanation (RKHS, Holder pairs, $\mathcal{I}$-characteristic kernel, ghost samples). Additionally, it is unclear if/when $m$ independent trajectories are used in the theoretical/experimental computation of MMD. Should $\kappa$ depend on $m$ instead of $n$?
- Organization: Some sections such as 4.3, 5.2, and 7.3 are included for completeness, but are not critical for the theoretical results or experiments. It may be better to move them to the Appendix to improve the paper's flow.

---

> ### Author Response · Authors · 2022-10-05
> **Reply to Review of Paper277 by Reviewer bTrg Part 1**
>
> We want to thank the reviewer for the insightful comments and thorogh review.
>
> - kernels
>
> Thank you pointing out this important aspect. In this paper we propose an extension of the well-established two-sample test of Gretton et al. to dynamical systems. To focus on our core contribution, we adopted all hyperparameter settings from the ones originally presented by Gretton et al. i.e., we use the squared exponential kernel and the median heuristic to obtain its length scale.
>
> We have added a sentence to the manuscript in the beginning of Sec. 7: "Further, following Gretton et al., we consider squared exponential kernels and chose the bandwidth based on the well-established median heuristic."
>
>  While in our experience, this setting is very robust, it is of course possible to further optimize parameters, see, for instance, "Optimal kernel choice for large-scale two-sample tests" by Gretton et al., which allows to reduce the Type II error.
>
>  We have also added the following paragraph to the paper:
>
> "Hyperparameter optimization can further improve the power of the test. We excluded such a discussion since this is not the main emphasize of our work and orthogonal to our contributions -- how can kernel two-sample tests be adapted to dynamical systems and how can a suitable notion of mixing be estimated from data."
>
> - Classification metrics
>
> Thank you for this suggestion. It was very insightful for us to reconsider what our mail goal is. The purpose of our paper is not to introduce a new classifier. Instead we show how kernel two-sample testing and MMD-based tests can be extended to dynamical systems. To demonstrate the importance and applicability of this extension, we use classification of gait data as an illustrative example.
>
> Nevertheless, we now calculated the per-class accuracy of our classification on the gait data:  No orthosis: $99.9 \%$ and  Orthosis: $91.47 \%$.
>
> The way our classification algorithm is designed here, precision, recall, and AUC are not suitable evaluation metrics. For these to be applicable, we require a decision boundary that is gradually increased. Instead, we fix a trajectory that we want to label. Next, we compare the trajectory against all known labeled trajectories by computing the MMD for each pair. Since the MMD yields a metric on the space of probability distributions, we pick the closest trajectory to determine the label.
>
> This way we leverage that the MMD is a metric to demonstrate how more complicated problems can be approached. In particular, whenever there is substantial deviation within each class. Then we are technically considering probability distributions over probability distributions. However, the illustrated approach works well.
>
> Alternatively, we could correct the test threshold and retrieve all trajectories that are below the threshold. This is particularly relevant when there are mistakes in the labels. Here, however, we are certain that all trajectories are labeled correctly.
>
> - One vs multiple trajectories
>
> There are two components to our approach:
>
> a) Estimating mixing times, and b) kernel two-sample testing.
>
> The ultimate goal of our approach is to compare two long trajectories via kernel two-sample testing. However, classical kernel two-sample testing is designed for i.i.d. data while data obtained from dynamical systems is highly correlated. Therefore, we we propose to first subsample from both trajectories such that the subsampled data is approximately i.i.d.
>
> The appropriate time shift for subsampling is determined through additional access to $m$ independent trajectories that we use to estimate the mixing speed. This yields a).
>
> With the estimated time shift we proceed to tackle b). We extract $n$ points from each of the two long trajectories. Due to the estimated time shift we can now apply kernel two-sample testing, which yields b).
>
> In Experiment 8.2.1, we have $35$ subjects and access to two different trajectories for each subject. One with an orthosis restricting the movement of the knee and one without the orthosis. Thus, we consider $m = 70$. Implicitly, we assume that applying an orthosis does not change the mixing speed of the dynamical system.
>
> For the presented experiments and in particular, the setting presented in Experiment 8.2.1, we apply the approach from Sec. 5.1, which has proven to be very robust.

---

> > ### Author Response · Authors · 2022-10-05
> > **Reply to Review of Paper277 by Reviewer bTrg Part 2**
> >
> > - Baselines
> >
> > The main goal of our paper is to unleash the standard kernel-two sample test onto dynamical systems. The gait data and classification example were chosen to demonstrate: a) the relevance of dynamical systems, b) that our new mixing assumption is suitable for real-world problems, and c) that subsampling via the estimated mixing speed allows us to treat the data as independent and thus, we can use any down stream classifier.
> >
> > The presented example proves all three points. We compare to reasonable baselines that are used in practice. In general, the performance of statistical tests for dynamical systems heavily depends on the system.
> >
> > Apart from the two-sample test problem, we provide a new method to analyze the mixing behavior, which is a critical component when dealing with dynamical systems.
> >
> > - Additional experiments
> >
> > We would refer you to our answer to the last question of reviewer SL5a.
> >
> > For your convenience, we have copied our answer here:
> >
> > We are not claiming that we propose the single best method for time series classification and further, strongly believe that depending on the specific system properties and problem requirements, different algorithms thrive.
> > Ours is designed for problems, where we have access to many independent and long trajectories.
> > For this type of systems, our method performs very well.
> > We discuss in detail the necessary assumptions and we show what can go wrong for appropriate systems that fit in this problem class. Further, we validate our method on real-world data and show that it performs well for a suitable system.
> >
> > Thank you for pointing us at the excellent data set.
> > We have analyzed the earthquake data since it was the first sensor-based dataset that consists of two labels. Interestingly, the data, which is labeled as no major event does not mix. When considering the other set of trajectories, we obtain fast mixing behavior. Intuitively, it makes sense that active earthquakes cause strong movements and thus, mixing. For the other trajectories, there is less movement. It would be interesting to investigate the non mixing class for longer time scales.
> > Moreover, this system is not stationary since at some point an earthquake appears.
> >
> > We have applied the kernel two-sample test without subsampling, even though the system is non-mixing and non-stationary.
> > In detail, we compute the pairwise MMDs between the test set and the train set. Afterward, we pick the trajectory with the smallest value and assign the label to the trajectory from the test set.
> > Additionally, we compute the mean over all the MMD values that are used for classification. There are some trajectories that look very different from all the others and show an extremely high MMD. These are therefore excluded and moved to the label unknown.
> >
> > Overall, we classify $51,08\%$ of the trajectories correctly, $22,30\%$ as unknown, and $26,62\%$ wrong. When looking at the two classes separately, we obtain for the "no major event" label the following results: $65,38$ correct, $22,12\%$ unknown, and $12,5\%$ wrong. For the trajectories labeled as earthquakes: $8,57\%$ are correct, $22,86\%$ unknown, and $68,57\%$ are wrong.
> >
> > The weaker performance of our test when compared to a nearest neighbor method can in parts already be explained by the non-stationarity especially of the data containing earthquakes. Further, we here propose a two-sample test, i.e., a method that is supposed to decide whether two samples are drawn from the same distribution. Such a method should only be used for classification if it can reasonably be assumed that samples with the same label follow the same distribution while samples with different labels come from different distributions. For the earthquake dataset, this is not a reasonable assumption. The labels are determined by a value on the Richter scale, where everything above 4 is classified as an earthquake. However, trajectories with different values on the Richter scale, even if all of them are either above or below 4, may very well follow completely different distributions.
> >
> > Because of the problems of this data and because we do not think it would add substantial insight to the paper compared to the other results, we decided to not include the datasets. We will keep this for future investigations.  Thanks again for pointing us to the dataset.
> >
> > - Noisy measurements
> >
> > Thank you for this insightful comment. We have considered it and decided to not include an extra example for this problem. The paper is already quite long and this would take up even more space while only adding little new insights.

---

> > > ### Author Response · Authors · 2022-10-05
> > > **Reply to Review of Paper277 by Reviewer bTrg Part 3**
> > >
> > > - $k$-nearest neighbor
> > >
> > > The gait data and classification example was chosen to demonstrate: a) the relevance of dynamical systems, b) that our new mixing assumption is suitable for real-world problems, and c) that subsampling via the estimated mixing speed allows us to treat the data as independent and thus, we can use any down stream classifier.
> > >
> > > Our goal was to demonstrate an interesting and nontrivial use case. Our baseline shows that the problem is nontrivial and the applied kernel two-sample test approach illustrates the power of kernel tests for dynamical systems even though there is room for improvement.
> > >
> > > We have realized that our wording in the paper might have been too strong and we weakened some formulations.
> > >
> > > "The above results show that the proposed method works well on a practically relevant non-trivial problem and outperforms common baselines."
> > >
> > > Changed to
> > >
> > > "The above results show that the proposed method works well on a practically relevant non-trivial problem and beats reasonable baselines."
> > >
> > > Also, we have added a sentence to the conclusion where we emphasize for what class of systems our method is designed for:
> > > "The method is tailored to dynamical systems for which we have access to multiple independent and long trajectories."
> > >
> > > - Discrete-time to continuous-time
> > >
> > > For our approach, there is no difference if the underlying dynamical system is defined in continuous or in discrete time. It works for both cases. We analyze the underlying stationary distribution and thus, discretization or irregular sampling times are not an issue.
> > >
> > > For example, the presented Lorenz system is defined in continuous time and we obtain the data by solving the ODE and then sampling from it. In the end, data obtained from dynamical systems are typically measurements from a continuous-time systems sampled at discrete times.
> > >
> > > - Thank you for pointing out the typos, we have corrected them. Regarding Figure, Theorem, Appendix, etc. should be capitalized when referenced in the text: We very much agree with the reviewer. However, the TMLR style guidelines state the opposite, which is why we have chosen lower case letters in the manuscript.

---

### Review · Reviewer_8fMv · 2022-09-21

**Summary Of Contributions:**

This paper presents a two-sample test in the style of MMD-based kernel two sample tests that extends beyond the IID scenario, which most other tests assume. In particular, it focuses on the case of dynamical systems, were data is not IID but can be handled by defining a suitable notion of mixing. The key techincal hurdle is to define this notion of mixing; the paper does so using an MMD criterion between the point and a time-shifted one, which can extended into a notion of mixing that compares a trajectory and a shifted and slower-sampled one. With this notion of mixing in hand, the paper turns it into a two-sample test by leveraging known results from the IID setting. The paper presents empirical experiments on simple illustrative examples (Section 7) and a real application, of Human Walking data. The results seem to show that the test leads to reasonable classification performance.

**Broader Impact Concerns:**

* No broader impact concerns.

**Requested Changes:**

* Discuss in more detail how the time shift is selected in practice. The paper currently makes vague references to Def 7, which is not an algorithmic definition.
* Related to above. When one is 'not able to estimate an a*', as stated in page 12. What does this mean exactly? Please elaborate on how this is to be diagnosed.
* Inlcude quantiative metrics in Sections 7.1-7.3. I might be missing something, but if the main contribution of the paper is a two sample test, then the empirical validation should focus on the power and validity of such test. Yet, Section 7 focuses on the qualitative visual evaluation of the samples generated via time-shift. I would suggest bringing some results from the Appendix to the main text.
* Add comparison to baselines. It is hard to gauge the quality of this method without comparison to any relevant baseline - even naive application of iid two-sample tests, or (ideally) more advanced methods, e.g. two-sample tests for functional data (Wynne et al 2020),  or related.
* Add test power analysis. There is currently no discussion of the power of this test, which is suprising for a paper on two-sample testing. In particular I would expect to see power/Type II error evaluation results, like most of the cited papers on kernel two-sample tests include.
* Discuss in more detail the relation to (Chérief-Abdellatif & Alquier, 2022) and (Wynne and Duncan, "A Kernel Two-Sample Test for Functional Data, 2020 -- not cited here).
* Clarify the following points:
    * How can Definition 3 imply ergodicity? It seems like something is missing here, e.g., that Eq (2) = 0 for that implication to hold.
    * Isn't assuming that the dynamical systems have converged a very strong assumption in Section 3.2?
    * Can you more clearly state why the sets in Pg 7 are "per construction" idd within themselves?
    * Prop 2 should make explicit that \kappa(n, \alpha) is defined above.
    * I don't understand what "respecting the estimated time shift" means in Page 10.

**Strengths And Weaknesses:**

Strenghts:
* A clever method to extend existing kernel two-sample tests from the IID to the non-IID dyanmic setting
* The paper is overall well written

Weaknesses:
* The core of the method is a somewhat pedestrian modification (time-shifting) to standard tests for iid data.
* The paper is missing some important details on a crucial component of this method - the choice of time shift.
* The experiments lack details, comparison to any baseline, and thorough evaluation.
* Comparison to some related work is very sparse

---

> ### Author Response · Authors · 2022-10-05
> **Reply to Review of Paper277 by Reviewer 8fMv Part 1**
>
> First of all, we want to thank the reviewer for the helpful comments and pointing us at more related work that helped us improve our paper.
>
> - Time shift
>
> If we assume that a system is mixing, then there exists a time shift $a^* < \infty$ that guarantees that for all $a'>a^*$ the dependency stays sufficiently small.
>
> In practice, we propose to pick the first time shift that yields an HSIC that is smaller than the test threshold. Clearly, we cannot guarantee that it will not increase again afterward. Nevertheless, data that is sampled according to this shift exhibits (almost) independent samples from the stationary distribution and hence, is suitable for subsequent kernel two-sample testing. In A.1.1, we discuss and describe the choice of $a^*$ in more detail.
>
> We have added an additional sentence to the main corpus of the manuscript: "Thus, we estimate $a^*$ based on MMD-mixing (cf. Def. 7) and (unless stated otherwise) pick the first time shift that is below the test threshold."
>
> - Follow up on time shift
>
> Given a finite length of the trajectory, it is easily possible that the first and the last point are still heavily correlated. That is, even by picking the first and last point and thus, enforcing the maximal possible time shift, we do not obtain sufficiently mixed data. An intuitive example are LTI systems $x_{k+1} = Ax_k + \epsilon_k$, where dynamics $A$ are close to the identity matrix and $\epsilon$ has zero mean and tiny variance.
>
> If we apply the HSIC in such cases, we naturally reject the null hypothesis that data is independent. The diagnosis of this would be that the even the largest time shift is not sufficient for the system to mix sufficiently and, hence, the kernel two-sample test should not be applied. This is a strength of our work. Rather than assuming mixing beforehand, it is estimated from data .
>
> In our numerical experiments, we have randomized the dynamics and noise matrix. By sampling often enough, we will also pick systems that mix slowly. As described above, we could detect the slow mixing properties. For the numerical examples, we rerolled the dynamics if the time shift exceeded an a priori defined maximal value.
>
> Based on your comment, we have reworded the corresponding statement in the manuscript to make this point more explicit: "With the proposed method, we would notice this since we would not be able to estimate an appropriate $a^*$ due to substantial remaining correlations in the data."
>
> - Quantitative metrics
>
> The main contribution of our work is showing how dynamical systems can be compared via kernel two-sample testing. We propose a data-driven approach to estimate mixing properties from data and show how correlations decay over time. In particular, we show how to chose suitable time shifts that yield nearly independent data. After subsampling the trajectories, we are comparing i.i.d. samples drawn from the stationary distribution. The power of kernel two-sample tests for i.i.d. data has extensively been discussed by Gretton et al.
>
> - Baselines
>
> We strongly believe that depending on the dynamical system and specific problem requirements, different methods are advantageous and there is no single best method for analyzing dynamical systems. Our method thrives for stationary systems and long trajectories and allows to generalize many state-of-the-art methods that require i.i.d. assumptions directly to dynamical systems. This valuable suggestion of the reviewers made us realize that we should stress this point in the paper as well and we we have included an additional sentence in the conclusion section:
> "The method is tailored to dynamical systems for which we have access to multiple independent and long trajectories.
>
> At the heart of our approach, we compare the underlying stationary distribution. Other methods (for example, (Wynne \& Duncan, 2022)) compare the joint distributions of trajectories. We have extended the related work section where we discuss connections to (Wynne \& Duncan, 2022) and (Chérief-Abdellatif \& Alquier, 2022). In our answer to the next question, we discuss this point in more detail.
>
> The MMD is a well-established and widely used method for comparing i.i.d. data. Its general quality has been demonstrated in various problem settings. However, it is not applicable to the large space of dynamical systems whose data is non-i.i.d. and therefore, technical assumptions are necessary. In particular, assumptions on mixing times. We circumvent this pitfall by estimating mixing times from data. There is no other method that estimates mixing times from data.
>
> Additionally, our method is advantageous from a computational point of view since we do not require sophisticated kernels and aim for maximizing the information content of each data point. Thus, there are many potential applications, involving embedded and battery powered devices that we envision as application areas.

---

> > ### Author Response · Authors · 2022-10-05
> > **Reply to Review of Paper277 by Reviewer 8fMv Part 2**
> >
> > - Related Work
> >
> > Thank you for pointing out the paper (Wynne and Duncan, "A Kernel Two-Sample Test for Functional Data, 2020).
> > We have included it in our discussion on related work and expanded the discussion on (Chérief-Abdellatif $\&$ Alquier, 2022).
> >
> > "[...] similar idea of mixing in RKHS, has very recently been introduced in (Chérief-Abdellatif \& Alquier, 2022). The paper, however, focuses on parameter estimation with respect to minimizing the MMD as a loss function. Further, the precise notion of mixing differs from ours. It is shown that certain types of systems satisfy their notion of mixing, however, that work does not estimate mixing from data. In (Wynne \& Duncan, 2022), another recent approach is presented. In essence, the paper investigates whether two samples of functions have the same underlying distribution. Functional data is directly embedded into an RKHS by extending the theory to kernels that live on function spaces. The elegant kernel design might be a useful for extensions of our work, where multiple correlated joint distributions need be compared. Right now, we compare stationary distributions and data is decorrelated via mixing."
> >
> > The elegant kernel design from (Wynne \& Duncan, 2022) can potentially be combined with our work to compare multiple correlated samples from joint distributions. In our opinion, it is not a competitor method but instead might help us to generalize our method.
> >
> > - Ergodicity
> >
> > Thank you for pointing out this mistake. You are correct and we have adjusted the sentence accordingly. For mixing, Eq (2) has to be zero, which then implies ergodicity.
> >
> > - Convergence
> >
> > We would argue that it is a very reasonable assumption to model a large class of highly relevant system. For example, bio-medical systems such as human walking or a beating heart. They exhibit a stationary and repetitive behavior while showing enough stochasticity or chaos to also mix.
> >
> > Similar arguments hold for engineering systems that include feedback control and, hence, converge to a stationary regime but still exhibit some (random) movement, e.g. due to noise. Another example are systems with a repetitive motion such as pumps.
> >
> > Of course, we also exclude certain types of systems and problems due to the mentioned assumption. However, we strongly believe that we strike a good balance between necessary mathematical assumptions in order to obtain control over system, while still covering a large class of potential and relevant applications.
> > We have added a sentence to the conclusion section (as mentioned in our answer to your Comment 4).
> >
> > - "per construction" idd
> >
> > We assume access to $m$ independently recorded trajectories from the same system. Due to the stationarity and ergodicity assumptions, all $X_0^{(i)}$ are drawn independently from the same distribution. Same holds for all $X_a^{(i)}$.
> >
> > We have adjusted the sentence to make this more explicit. "The sets are, per construction, i.i.d. within themselves since the $m$ trajectories are independent and due to the additional stationarity and ergodicity assumptions."
> >
> > - kappa
> >
> > We have added a corresponding sentence to Proposition 2.
> > "The choice of the threshold $\kappa (n, \alpha)$ is discussed extensively in Gretton et al. (2012a) and also above."
> >
> > - "respecting the estimated time shift"
> >
> > It means that data is time shifted by the estimated $a^*$ in contrast to time shifts that are deliberately chosen smaller to illustrate what can go wrong.

---

### Decision · Action_Editors · 2022-12-01

**Recommendation:** Reject

**Comment:**

From a novelty point of view, the method is a very simple extension of vanilla MMD and the theoretical contributions largely rest of vanilla MMD results. That by itself is not a short-coming, rather it is a strength of the proposed approach. At the same time, the value of the paper resides equally in document a convincing demonstration of the effectiveness of the approach - which includes guidance to practitioners on how best to choose kernels, hyper-parameters etc and how sensitive the tests are to those choices; how the method compares with other natural baselines; and how it enables new applications. The reviews capture how the paper falls a bit short on this front, pointing out the criticality of having "a better experimental evaluation ... strong baselines with learned feature representations, such as another kernel method or a neural network". The authors are encouraged to resubmit after taking these comments into account.




**Audience:**

The proposed methods have clear applications in Robotics for novelty detection / change detection applications; or biomedical applications such as the one considered in this paper where changes in human sensor data (walking gaits) could signal disease conditions. Hence, the themes of this paper would definitely be of interest to several communities.

**Claims And Evidence:**

Inducing whether two distributions are identical or not by computing the maximum mean discrepancy metric over their associated RKHS embeddings is a very elegant approach, originally designed for i.i.d random variables. It is natural to ask how such a framework can be extended to non-i.i.d trajectory data generated by two dynamical systems. This paper ultimately proposes a simple method, which is to apply MMD on trajectory snippets sufficiently separated in time, using suitable tensor kernels.  All reviewers are largely in agreement that the method is simple, technically correct, and well presented. The main concerns are of the flavor that the method is perhaps an underwhelmingly simple modification (time-shifting) to the vanilla MMD approach; and that the experimental results could provide more  details, comparisons to baselines, and more thorough evaluation. Concerning the former issue, the paper makes useful contributions to provide justifications for theoretical soundness of the tests.

The latter issue does seem to be a significant concern: the method is shown on some analytical dynamical systems with parameter perturbations; and a couple of comparisons are reported on human walking data. An analysis of choice of kernel and its hyper-parameters is missing.  . It is hard to gauge the quality of this method without comparison to more comprehensive baselines - as the reviews point out even naive application of iid two-sample tests, or (ideally) more advanced methods, e.g. two-sample tests for functional data.